# AdaRank: Adaptive Rank Pruning for Enhanced Model Merging

**Chanhyuk Lee, Jiho Choi, Chanryeol Lee, Donggyun Kim and Seunghoon Hong**
KAIST
`{chan3684, jiho.choi, lcy9442, kdgyun425,`
`seunghoon.hong}@kaist.ac.kr`

## Abstract

Model merging has emerged as a promising approach for unifying independently fine-tuned models into an integrated framework, significantly enhancing computational efficiency in multi-task learning. Recently, several SVD-based techniques have been introduced to exploit low-rank structures for enhanced merging, but their reliance on heuristically designed rank selection often leads to inter-task interference and suboptimal performance. In this paper, we propose **AdaRank**, a model merging framework that replaces this heuristic selection by adaptively selecting the beneficial singular components of task vectors to merge multiple models. We first show empirically that (i) selecting only the top singular components of task vectors can cause critical interference with other tasks, and (ii) assigning fixed ranks does not align with the varying complexity of tasks and layers. AdaRank addresses both issues by adapting per-component masks, indicating the selection of the component, to the unlabeled test data with entropy minimization. Our experimental findings show that AdaRank consistently improves existing merging methods across diverse backbones from different modalities, largely narrowing the performance gap against individually fine-tuned models.

## 1 Introduction

Recent advancements in machine learning have significantly enhanced the performance and diversity of pre-trained models, enabling the widespread availability of fine-tuned models tailored to specific tasks across various domains (Dosovitskiy et al., 2020; Devlin et al., 2019). These developments have made high-quality, task-specialized models increasingly accessible, often distributed through public repositories for easy deployment. However, utilizing each fine-tuned model independently remains computationally expensive and impractical, particularly as the number of tasks grows in real-world applications. Consequently, *model merging* (Li et al., 2023) has emerged as a promising solution to integrate individually fine-tuned models into a unified framework, facilitating scalable multi-task performance without requiring extensive retraining or resource-heavy infrastructure.

Among the pioneering techniques in model merging, Task Arithmetic (Ilharco et al., 2023) combines task vectors—defined as the difference between fine-tuned and pre-trained model weights—to integrate multiple models into a single framework, enhancing multi-task performance without requiring access to train data. Building on this baseline, several studies have proposed methods to modify these task vectors in an element-wise manner (Yadav et al., 2023; Huang et al., 2024; Wang et al., 2024) to address inter-task interference, a key factor contributing to the performance gap between merged and fine-tuned models. More recently, researchers have adopted Singular Value Decomposition (SVD) to adjust task vectors in the spectral domain rather than through element-wise modifications (Choi et al., 2024; Lu et al., 2024; Lee et al., 2025; Gargiulo et al., 2025), reporting improved performance.

Despite these advances, SVD-based methods still fail to fully close the performance gap with fine-tuned models, primarily due to their heuristic selection of the low-rank subspace. Specifically, we identify two important behaviors of SVD when applied to task vectors. First, incorporating singular components with large singular values often introduces greater inter-task interference than if those components were removed. Second, the rank requirements corresponding to the complexity of the task vary substantially across tasks and layers. These observations imply that the commonly

used top-$k$ low-rank approximation in SVD-based merging cannot adequately mitigate inter-task interference and may lead to suboptimal performance.

To address these challenges, we propose **AdaRank** (**Ada**ptive **Rank** Pruning), a method that replaces the rigid top-$k$ heuristic with a dynamic selection of singular components. The core of AdaRank is a set of learnable binary masks, one for each singular component of every task vector, which determines whether a component should be pruned or preserved. This formulation enables finding a set of singular components that introduces minimal inter-task interference. To navigate this process without measuring the multi-task loss, AdaRank employs test-time adaptation (Sun et al., 2020; Wang et al., 2021), using Shannon entropy minimization as an unsupervised objective, inspired by AdaMerging (Yang et al., 2023).

We evaluate the effectiveness of AdaRank across diverse backbones and varying numbers of tasks, including both vision and language transformers, and show that it significantly narrows the performance gap with individually fine-tuned models. AdaRank also integrates seamlessly with different families of model merging strategies—static approaches (Ilharco et al., 2023; Choi et al., 2024; Gargiulo et al., 2025), adaptive methods (Yang et al., 2023), and post-calibration techniques (Yang et al., 2024a)—independently improving their performance while preserving their respective advantages. Moreover, we demonstrate that our method achieves performance on par with or better than router-based adaptive approaches (Lu et al., 2024; Tang et al., 2024b), despite requiring no additional parameters, highlighting the efficiency of our method and underscoring its potential as a versatile solution for model merging.

## 2 PRELIMINARIES

**Task Arithmetic.** Given $T$ heterogeneous tasks and model parameters $\theta_i$ for $i = 1, ..., T$ fine-tuned from the same pre-trained backbone $\theta_0$, model merging aims to build a merged parameter $\theta_m$ capable of performing all tasks. We consider the layer-wise Task Arithmetic (TA) (Ilharco et al., 2023) as a base approach that obtains the merged parameter by:

$$\theta_m^l = \theta_0^l + \lambda^l \sum_{i=1}^{T} \tau_i^l, \tag{1}$$

where $l$ denotes the layer index, $\theta^l \in \mathbb{R}^{d \times d'}$ is the parameter of $l$th layer, $\lambda^l \in \mathbb{R}$ is the layer-wise scalar coefficient, and $\tau_i^l = \theta_i^l - \theta_0^l$ is a *task vector*[1] representing task-specific knowledge encoded in the difference between the fine-tuned and pre-trained parameters.

While TA has been effective in various model merging scenarios, it has also been widely observed that its merging performance is largely limited by inter-task interference in task vectors *i.e.,* adding a task vector degrades the performance of the other tasks. To address the issue, various attempts have been made to truncate the conflicting components of the task vector.

**Task Vector Truncation.** Early approaches propose to truncate task vectors based on their *element-wise* contribution to the merged model. A core technique in these works can be expressed as multiplying an element-wise mask to the task vector by $\hat{\tau}_i = M_i \odot \tau_i$, which are carefully designed to reduce conflicts across task vectors (Yadav et al., 2023; Yu et al., 2024; Huang et al., 2024). However, such hand-designed, per-element pruning may fail to respect the inherent low-dimensional structure of the fine-tuned parameters, which stores critical knowledge for individual tasks (Denil et al., 2013; Denton et al., 2014; Hu et al., 2022).

Recent studies have discovered that exploiting the low-rank structure of task vectors can be an effective alternative (Choi et al., 2024; Gargiulo et al., 2025; Lee et al., 2025), often surpassing element-wise pruning. Under this framework, the model merging in Equation 1 is modified by:

$$\theta_m^l = \theta_0^l + \lambda^l \sum_{i=1}^{T} \text{SVD}_k(\tau_i^l), \tag{2}$$

---

[1]In layer-wise approaches, each task vector $\tau^l \in \mathbb{R}^{d \times d'}$ is actually a matrix but we follow the convention of calling it a vector (Ilharco et al., 2023).

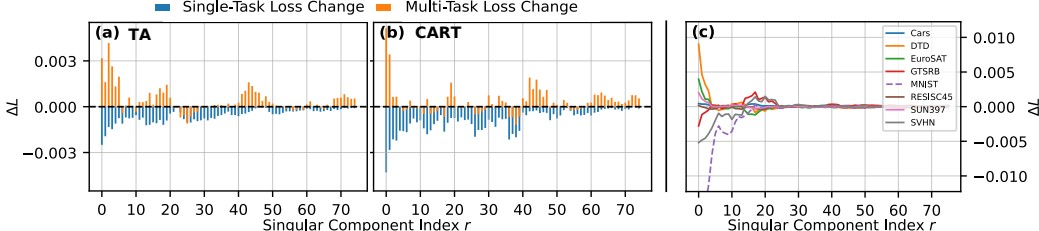

Figure 1: **(a), (b)** Net change in single-task and multi-task losses for Task Arithmetic (Ilharco et al., 2023) and CART (Choi et al., 2024), respectively, when each singular component of a target task vector is individually added to a model merged with full-rank vectors from other tasks. **(c)** Loss changes of all tasks when adding singular components from the MNIST task vector, with MNIST shown as a dotted line. For clarity, only the top 10% of singular components are shown.

where $\text{SVD}_k$ denotes the Singular Value Decomposition with low-rank approximation on top-$k$ singular components. Unlike element-wise modification, truncating the singular components of the task vector retains the core low-dimensional structure of the original matrix, thereby preserving the rich information of individually fine-tuned task vectors. Meanwhile, maintaining the rank of task vectors $k$ small helps to reduce the potential of inter-task interference across tasks (Lee et al., 2025).

Despite their success, the inter-task interference is still inherent in SVD-based methods. This is primarily because each task's singular components are obtained via independent SVD per task, allowing correlations to persist in singular components across tasks. While the prior works introduced additional mechanisms to further ensure orthogonality of task vectors via whitening (Gargiulo et al., 2025) or reorientation (Choi et al., 2024), it does not fully address the issue, as their merging performance highly depends on the choice of rank $k$.

## 3 ANALYSIS ON SVD OF TASK VECTORS

Building on the low-rank paradigm, we seek an alternative direction for enhancing model merging. Specifically, we scrutinize the common practice of truncating each task vector to its top-$k$ singular components, and pose two central questions: **(1)** *Is choosing the top singular components always beneficial for model merging?* **(2)** *Is enforcing a fixed rank across tasks and layers desirable for model merging?* In this section, we conduct an empirical analysis to investigate these questions, revealing the limitations of naive top-$k$ truncation.

**Limitations of Top Singular Components.** While it is guaranteed that the top-$k$ singular components yield the best low-rank approximation in terms of reconstruction error for a single matrix (Eckart & Young, 1936), this optimality does not necessarily transfer to multi-task model merging. In a single-task scenario, these top components—characterized by large singular values—effectively minimize the loss for that particular task. However, in a multi-task setting, these top components can also exhibit significant interference across tasks, potentially degrading the merging performance.

To measure the effect of adding singular components under other tasks' interference, we set up the following experiment: for task $i$, we construct a merged model $\theta_m$ using full-rank task vectors of the other tasks $j \in \{1, ..., T\} \setminus \{i\}$. Thus, $\theta_m$ is defined as

$$\theta_m = \theta_0 + \lambda \sum_{j \neq i} \tau_j. \tag{3}$$

Then, to isolate the impact of each singular component, we measure the change in total multi-task loss when adding only the $r$-th singular component (enumerated in descending singular value order) of the $i$-th task $\mathbf{s}_{ir} = \sigma_{ir} \mathbf{u}_{ir} \mathbf{v}_{ir}^\top$ to $\theta_m$. We repeat this for each $r$ by:

$$\Delta L(r) = \sum_{t=1}^{T} \Delta L_t(r) = \sum_{t=1}^{T} L_t(\theta_m + \lambda \mathbf{s}_{ir}) - L_t(\theta_m). \tag{4}$$

Figure 1 **(a)** summarizes the results. It shows that top-singular components tend to contribute more in reducing the loss for its own task $\Delta L_i$ (blue bars), while also introducing significant interference

to the other tasks, often resulting in a net increase in multi-task loss $\Delta L$ (orange bars). It indicates that naively merging top singular components can often be suboptimal in multi-task model merging, since the interference with other tasks may outweigh the performance gains from adding the top singular component of a single task. This trend is not specific to one method; we observe a consistent pattern with CART (Choi et al., 2024), which defines task vectors as the deviation between fine-tuned weights and their average, as shown in Figure 1 **(b)**.

For a deeper understanding of this phenomenon, we analyze the loss changes of individual tasks exemplified by the case where the excluded task $i$ is MNIST (LeCun, 1998) (Figure 1 **(c)**). We observe that adding a singular component from the MNIST benefits semantically aligned tasks (e.g., SVHN (Netzer et al., 2011), digit classification task), whereas dissimilar tasks (*e.g.*, DTD (Cimpoi et al., 2014), a texture classification task) experience increases in net loss. These varying interactions lead to a complex interplay of losses for each singular component. In particular, top components with large singular values $\sigma_{ir}$ can have a more pronounced negative impact on dissimilar tasks, increasing their net loss. Overall, these results suggest that top singular components are not always optimal for multi-task model merging. For additional extensive analysis on the singular components of task vectors, see Appendix C.1.

**Limitations of Fixed Rank Truncation.** Another significant limitation lies in the use of a fixed top-$k$ truncation across diverse tasks and model layers. In Figure 2, we quantify the intrinsic rank of the task vectors by measuring the number of singular components required to preserve 95% of the total variance, or spectral energy (i.e., the sum of squared singular values) (Jolliffe, 2011; Raschka & Mirjalili, 2019), which captures the effective linear dimensionality of the matrix. The intrinsic rank varies considerably across task vectors from different models. Consistent with the finding that the intrinsic dimension of neural network representations increases with task complexity (Li et al., 2018), our experiment supports that this trend

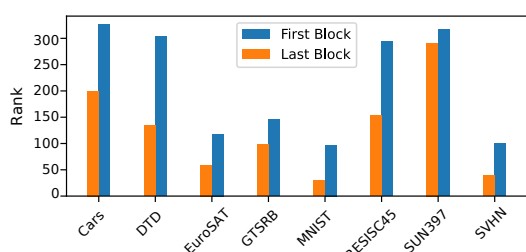

Figure 2: Intrinsic rank capturing 95% of total spectral energy in the MLP layer of the first and the last block of ViT-B/32 task vectors obtained from 8 different fine-tuned weights.

is also observed in task vectors, revealing a strong correlation between intrinsic rank and task demands. For instance, task vectors from models fine-tuned with SUN397 (Xiao et al., 2016) (397 classes) require a higher rank compared to those from simpler tasks like MNIST (LeCun, 1998) or SVHN (Netzer et al., 2011). Moreover, task vectors exhibit pronounced rank variation across layers (see Figure 2). Early layers, which capture task-agnostic features (Krizhevsky et al., 2012; Raghu et al., 2017; Papyan et al., 2020), show higher ranks with lower variance, reflecting shared information across tasks. In contrast, later layers, encoding task-specific representations, demonstrate greater rank variability and lower overall ranks. This divergence of rank among tasks and layers highlights the challenge of using a fixed top-$k$ truncation for model merging, since we may either discard important components that are critical for some tasks or keep unnecessary components that cause interference between tasks.

**Summary.** Summarizing these observations: **(1)** Some of the top singular components benefit their own tasks but degrade performance in other tasks, making naive top-$k$ selection suboptimal on multi-task performance. **(2)** Task vectors exhibit diverse rank requirements across tasks and layers, so a fixed rank truncation fails to accommodate these differences. These findings highlight the necessity of an *adaptive* selection strategy that can *selectively preserve* critical singular components for each task and layer while mitigating negative interference.

## 4 ADAPTIVE RANK PRUNING

In this section, we introduce **AdaRank**, **Ada**ptive **Rank** Pruning, a test-time adaptation (Sun et al., 2020; Wang et al., 2021) method to find the advantageous subset of singular components that minimize the inter-task interference in model merging, while preserving each task's performance.

**Selective Pruning with Binary Masks.** To selectively preserve or prune singular components, we introduce a binary mask that indicates the selection of each component. For each layer $l$ and task $i$, we define a binary mask vector $B_i^l \in \{0, 1\}^{1 \times m}$ ($m = \min(d, d')$ for $\tau^l \in \mathbb{R}^{d \times d'}$), where each element indicates whether the corresponding singular component is preserved (1) or pruned (0). The set of binary masks for all tasks in layer $l$ is denoted as $B^l = \{B_1^l, B_2^l, \ldots, B_T^l\}$. For brevity, we denote the collection of all such masks across layers and tasks simply as $B$. Under a given state of $B$, the layer-wise merged model is expressed as:

$$\theta^l(B^l) = \theta_0^l + \lambda^l \sum_{i=1}^{T} U_i^l(\mathrm{diag}(B_i^l) \odot \Sigma_i^l)V_i^{l\top}, \tag{5}$$

where $U_i^l$ and $V_i^l$ are the left and right singular vector matrices, $\Sigma_i^l$ is the diagonal matrix of singular values, and $\odot$ denotes the element-wise product.

Note that when $B_{ir} = 1$ for $r \leq k$ and $B_{ir} = 0$ for $r > k$ for all $i$, Equation 5 reduces to the top-$k$ selection strategy (Equation 2). On the other hand, if all elements of $B$ are set to one, it reduces to standard Task Arithmetic composed of full-rank task vectors (Equation 1). Allowing $B$ to be a set of arbitrary binary vectors, we can express any combination of singular components of each task vector. This effectively addresses issues of naive top-$k$ truncation discussed in Section 3: we can selectively prune or preserve the singular components according to their effect on inter-task interference, and allow variable ranks in task vectors across tasks and layers.

**Test-Time Mask Adaptation.** Having established a flexible selection mechanism, the critical question becomes how to determine the configuration of the masks. Ideally, this configuration could be optimized to minimize the net multi-task loss, the most direct measure of overall task interference. However, we cannot access training data or directly compute the gradient of supervised loss during model merging. Instead, we adopt test-time adaptation with Shannon Entropy (Shannon, 1948) minimization as a surrogate objective for optimizing $B$ without access to training data or labels of test data. Entropy minimization has been widely employed as a surrogate objective in dynamic variants of model merging (Yang et al., 2023; 2024a; Lu et al., 2024), and it has generally proven effective due to its strong correlation with supervised multi-task loss (Yang et al., 2023). Finally, our learning objective is minimizing the sum of output entropies $H_i$ for each task:

$$\underset{B}{\mathrm{argmin}} \sum_{i=1}^{T} \sum_{x_i \in \mathcal{D}_i} H_i(f(\theta(B), x_i)), \tag{6}$$

where $\mathcal{D}_i$ is a *unlabeled* test data for task $i$, and $f(\theta(B), x_i)$ is the model output parameterized by $\theta(B)$ for input $x_i \in \mathcal{D}_i$.

We optimize the binary values of $B$ using the Straight-Through Estimator (STE) (Bengio et al., 2013) with a sigmoid function, treating each entry of $B_i^l$ as a continuous parameter during the backward pass, consistent with prior works (Courbariaux et al., 2015; Hubara et al., 2016; Louizos et al., 2018). In the forward pass, entries of $B_i^l$ are rounded to $\{0, 1\}$ to serve as a binary mask, while in the backward pass, they remain continuous to propagate gradients. Once the final set $B$ is determined, we merge all the task vectors by applying each $B_i^l$ to Equation 5. The full algorithm of our framework and implementation of STE is explained in Appendix A.

## 5 RELATED WORKS

Before presenting experiments, we situate AdaRank within prior model merging research to clarify how our setting relates to these lines.

**Model Merging with Weight Sparsification.** To address inter-task interference of Task Arithmetic, several methods have been proposed to sparsify task vectors by removing redundant components in an element-wise manner. Notably, TIES-Merging (Yadav et al., 2023) selects dominant parameters from task vectors based on their magnitude and constructs a sign vector reflecting the prevailing sign across models. Consensus Merging (Wang et al., 2024) applies an additional mask to task vectors and extracts task-specific information through the $L_1$ norm difference, while DARE (Yu et al., 2024) randomly drops parameters and rescales the remaining ones. Despite the efforts, these methods still exhibit a noticeable performance gap against individually fine-tuned models.

**Model Merging in a Low-Rank Subspace.** Recent works leverage Singular Value Decomposition (SVD) to address interference between task vectors. CART (Choi et al., 2024) redefines task vectors as deviations from the average of fine-tuned weights and applies low-rank approximation to these task vectors. TSV-M (Gargiulo et al., 2025) proposes enforcing a low-rank structure on each task vector, followed by a whitening transformation to minimize interference among the truncated components. STAR (Lee et al., 2025) provides theoretical evidence that low-rank approximation of task vectors reduces the upper bound of interference in the merged model. While these SVD-based approaches narrow the performance gap to fine-tuned models, they still depend on heuristic rank truncation (typically fixed top-$k$), a limitation that our method directly addresses.

**Model Merging with Test-Time Adaptation.** A line of work adapts the merged model to the test data for further performance gain. The seminal AdaMerging (Yang et al., 2023) adapts task and layer-wise scaling coefficients via entropy-guided TTA, empirically showing that entropy loss could be an effective surrogate for supervised loss. Building on this, Representation Surgery (Yang et al., 2024a;b) suggests a *post-calibration* approach which adapts a lightweight MLP to align the merged model's representation of the test data with one from the fine-tuned model. Our method follows this line of work: rather than relying on heuristic top-$k$ truncation, we adapt our binary masks to select a low-rank subspace for each task vector.

**Router and Compression-Based Model Merging.** Router-based methods, also known as *Mo-Erging* (Yadav et al., 2024), retain per-task parameters and use a router module to combine them at each inference. WEMoE (Tang et al., 2024b) places per-block routers that dynamically select the MLP layers of task vectors, while Twin-Merging (Lu et al., 2024) adds a single router after the final layer to add low-rank task vectors to the shared weight. Alternatively, Compression-based methods such as EMR-Merging (Huang et al., 2024) and TALL-Masks (Wang et al., 2024) assume the task identity is provided at inference to select the task-specific parameters, instead of using the router. While effective, these methods require all task-specific parameters to remain accessible at inference, causing storage and memory to scale with the number of tasks. In contrast, our method retains no such parameters and deploys a model the same size as a single fine-tuned model.

# 6 EXPERIMENTS

## 6.1 EXPERIMENTAL SETUP

**Baselines.** We compare AdaRank with prior approaches in model merging, categorized into two groups: *static* merging methods, which do not include an adaptation process (*e.g.*, Task Arithmetic (Ilharco et al., 2023) and TIES-Merging (Yadav et al., 2023)), and *adaptive* merging methods, which incorporate a TTA process (*e.g.*, AdaMerging (Yang et al., 2023)), to adapt the merged model. For completeness, we discuss our method's compatibility with post-calibration methods (*e.g.* Representation Surgery (Yang et al., 2024a)) and performance comparison with compression-based methods (*e.g.* TALL-Masks (Wang et al., 2024)) in Appendix C.3.

**Implementation Details.** To demonstrate the generality of our method, we integrate AdaRank into two different baselines: Task Arithmetic (Ilharco et al., 2023) with SVD and CART (Choi et al., 2024), which correspond to Equation 2 with the difference in setting $\theta_0$ as a pre-trained model or weight averaging, respectively. For optimization, we initialize the binary matrix $B$ such that Equation 5 is initialized to Equation 1 for Task Arithmetic and to Equation 2 for CART. Additionally, we employ TSV-M (Gargiulo et al., 2025) and Iso-CTS (Marczak et al., 2025), another top-performing SVD-based merging method. For TSV-M, a whitening transform is applied to $U_i$ and $V_i$ matrices from Equation 5 (see Appendix A for details). Since the layer-wise coefficient $\lambda^l$ can be learned jointly with the binary mask $B$ via TTA as in AdaMerging (Yang et al., 2023), we learn both of them jointly in our default setting, with the effect from each $\lambda$ and $B$ is analyzed in Section 6.3. The specific initial values for $\lambda$ and $B$ are set according to the best-performing configurations described in their respective papers, with further details provided in Appendix B.1.

## 6.2 MAIN RESULTS

**Merging Vision Models.** Following standard experiment protocol (Wang et al., 2024), we evaluate our method by merging two Vision Transformer (Dosovitskiy et al., 2020) backbones of CLIP (Rad-

Table 1: Average accuracy along 8, 14, 20 vision tasks with merged ViT-B/32 and ViT-L/14.

| Method | ViT-B/32 | | | ViT-L/14 | | |
|---|---|---|---|---|---|---|
| | 8 tasks | 14 tasks | 20 tasks | 8 tasks | 14 tasks | 20 tasks |
| Pretrained | 48.0 | 59.6 | 56.0 | 65.0 | 68.4 | 65.4 |
| Individual | 90.5 | 90.3 | 90.5 | 94.2 | 93.3 | 94.0 |
| Weight Averaging | 65.9 | 64.3 | 60.9 | 79.6 | 76.8 | 71.7 |
| Task Arithmetic (TA) | 69.2 | 65.4 | 61.0 | 84.5 | 79.6 | 74.2 |
| Ties-Merging | 72.4 | 65.2 | 62.9 | 86.1 | 79.5 | 75.8 |
| Consensus Merging | 75.2 | 70.0 | 65.0 | 86.6 | 81.9 | 77.6 |
| TSV-M | 83.8 | 79.5 | 76.7 | 91.2 | 88.3 | 87.3 |
| CART | 84.7 | 79.5 | 76.8 | 92.6 | 88.0 | 87.9 |
| Iso-CTS | 84.9 | 80.6 | 77.2 | 93.0 | 90.2 | 89.6 |
| TA+AdaMerging | 80.1 | 76.7 | 69.2 | 90.8 | 88.0 | 86.8 |
| TA+**AdaRank** | **87.9** | **82.1** | **81.4** | **93.0** | **89.4** | **89.1** |
| TSV-M+AdaMerging | 87.1 | 84.5 | 83.5 | 93.0 | 90.6 | 91.7 |
| TSV-M+**AdaRank** | **88.9** | **86.9** | **86.9** | **93.7** | **92.1** | **92.8** |
| CART+AdaMerging | 85.9 | 82.3 | 82.7 | 93.1 | 90.4 | 91.3 |
| CART+**AdaRank** | **89.2** | **86.2** | **86.4** | **93.5** | **91.4** | **91.8** |
| Iso-CTS+AdaMerging | 89.2 | 85.4 | 84.7 | 95.4 | 92.5 | 92.1 |
| Iso-CTS+**AdaRank** | **89.4** | **86.8** | **86.5** | **95.5** | **93.4** | **93.3** |

ford et al., 2021): ViT-B/32 and ViT-L/14, which are fine-tuned on 8, 14, and 20 image classification tasks. The selected tasks cover a diverse range of domains and class counts, from fine-grained image classification on Flowers102 (Nilsback & Zisserman, 2008) to character recognition on KM-NIST (Clanuwat et al., 2018). We provide detailed dataset lists and descriptions in Appendix B.2.

The average accuracy of the merged model for 8, 14, 20 tasks and ViT-B/32, ViT-L/14 is presented in Table 1 (see Appendix C.4 for per-task breakdown). Applying AdaRank to static merging methods improves the final performance across all numbers of tasks and backbones. In particular, when applied to Task Arithmetic, it yields average gains of 18.6% and 11.1% for ViT-B/32 and ViT-L/14, respectively, outperforming all the best-performing static merging methods. Within adaptive methods, AdaRank significantly outperforms AdaMerging when applied to all three initial methods, which implies the effectiveness of adaptive selection of singular components compared to adapting only the task-wise coefficients. Moreover, we observe consistent gain by applying our method to different SVD-based merging frameworks using top-$k$ truncation (TSV-M $vs$ TSV-M+AdaRank, CART $vs$ CART+AdaRank). This improvement emphasizes that adaptive selection could offer further gain to the suboptimal performance of the rigid top-$k$ heuristic.

**Merging Language Models.** We further evaluate AdaRank on NLP tasks with RoBERTa-base (Liu et al., 2019) and GPT-2 (Radford et al., 2019) backbones. Following the setting from Fusionbench (Tang et al., 2024a), we merge 7 models fine-tuned on text classification tasks from the GLUE benchmark (Wang et al., 2018). Detailed descriptions are provided in Appendix B.2.

Results are presented in Table 2. Applying AdaRank in language models achieves a consistent gain in all the baseline methods. These findings confirm our method's effectiveness extends beyond vision models to different modalities. Furthermore, its success with both bidirectional (RoBERTa) and autoregressive (GPT-2) models demonstrates its architectural robustness.

Table 2: Average performance on 7 NLP tasks with merged RoBERTa and GPT-2.

| Method | RoBERTa | GPT-2 |
|---|---|---|
| Individual | 0.8483 | 0.7680 |
| Task Arithmetic (TA) | 0.6718 | 0.6064 |
| Ties-Merging | 0.6444 | 0.6112 |
| TSV-M | 0.6693 | 0.6195 |
| CART | 0.6997 | 0.6182 |
| TA+AdaMerging | 0.6762 | 0.5997 |
| TA+**AdaRank** | **0.7032** | **0.6328** |
| TSV-M+AdaMerging | 0.7065 | 0.6219 |
| TSV-M+**AdaRank** | **0.7309** | **0.6743** |
| CART+AdaMerging | 0.6954 | 0.6207 |
| CART+**AdaRank** | **0.7547** | **0.6587** |

Table 3: Average accuracy along 8, 14, 20 vision tasks with merged ViT-B/32 compared to router-based merging methods.

| Num.Tasks | 8 Tasks | 14 Tasks | 20 Tasks |
|---|---|---|---|
| Individual | 90.5 | 90.3 | 90.5 |
| TA+**AdaRank** | 87.9 | 82.1 | 81.4 |
| CART+**AdaRank** | 89.2 | 86.2 | 86.4 |
| TSV-M+**AdaRank** | 88.9 | 86.9 | **86.9** |
| Twin-Merging | 89.4 | 87.0 | 75.3 |
| WEMoE | **89.5** | **87.2** | 80.2 |

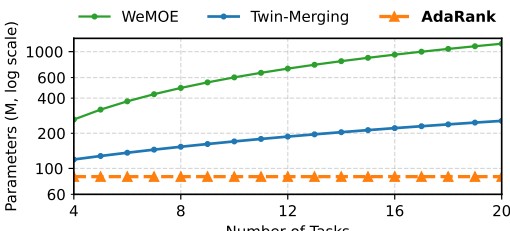

Figure 3: Parameter count of the merged model when merging ViT-B/32 finetuned on different number of tasks. We show y-axis with a log scale for better visualization.

Table 4: Number of learnable parameters and time consumption of AdaRank, compared to AdaMerging. We merge ViT-B/32 and ViT-L/14 fine-tuned on 8 tasks. The number of parameters is the total count along 8 tasks.

| Method | Total Params. | Learnable Params. | SVD Time | TTA Time | Performance |
|---|---|---|---|---|---|
| AdaMerging (ViT-B/32) | 907.6M | 2,440 | - | ∼10.3 min | 80.1% |
| **AdaRank (ViT-B/32)** | 907.6M | 294,912 (**0.032%**) | ∼0.3 min | ∼10.7 min | **87.9%** |
| AdaMerging (ViT-L/14) | 2.46B | 3,592 | - | ∼36.7 min | 90.8% |
| **AdaRank (ViT-L/14)** | 2.46B | 786,432 (**0.032%**) | ∼1.1 min | ∼37.2 min | **93.0%** |

**Comparison with Router-Based Methods.** We compare AdaRank with router-based methods, known as *MoErging* (Yadav et al., 2024), which retain task-specific parameters in non-merged form and train a router module to combine them according to each input at inference. We adapt all methods' learnable components on the same unlabeled data split via entropy minimization. Figure 3 and Table 3 show the parameter count and average performance for merging different numbers of ViT-B/32 models. While AdaRank maintains a constant size model (orange horizontal line), router-based methods scale linearly as they preserve all task-specific parameters and the router module (blue, green curve). In particular, merged models of WEMoE (Tang et al., 2024b) and Twin-Merging (Lu et al., 2024) are approximately 5× and 2.25× bigger than AdaRank for an 8-task benchmark, with these multiples increasing to 10× and 3×, respectively, for 20 tasks. Despite this disparity, AdaRank performs comparably when merging 8 and 14 tasks and even outperforms in a 20-task benchmark. These results highlight AdaRank as a more efficient alternative, achieving strong performance without the substantial parameter overhead inherent to router-based approaches.

**Adaptation-Time Cost.** Table 4 presents a breakdown of the computational costs of AdaRank in comparison to AdaMerging. In terms of learnable parameters, AdaRank introduces more variables because it must generate per-layer, per-task binary masks. Nevertheless, this overhead is marginal relative to the total parameter count, amounting to 0.032%. We also compare the wall-clock time for each singular value decomposition (SVD) and test-time adaptation (TTA). As shown in the table, AdaRank requires an additional single SVD step that is absent in AdaMerging; however, the time required is minimal compared to the total execution time. For the TTA phase, AdaRank consumes nearly the same time as AdaMerging. By efficiently utilizing the increased computational budget, AdaRank achieves significant performance gains of 7.8% and 2.2% on ViT-B/32 and ViT-L/14, respectively.

## 6.3 ANALYSIS AND ABLATION

In this section, we provide an empirical analysis to validate that AdaRank successfully resolves the two key limitations of the top-$k$ selection strategy identified in Section 3.

**Effect of Selecting Bottom Singular Components.** In Figure 4 **(a)**, we show the cumulated count of singular component indices selected by AdaRank, compared to fixed top-$k$. It indicates that AdaRank not only prunes the top singular components but also frequently selects the bottom com-

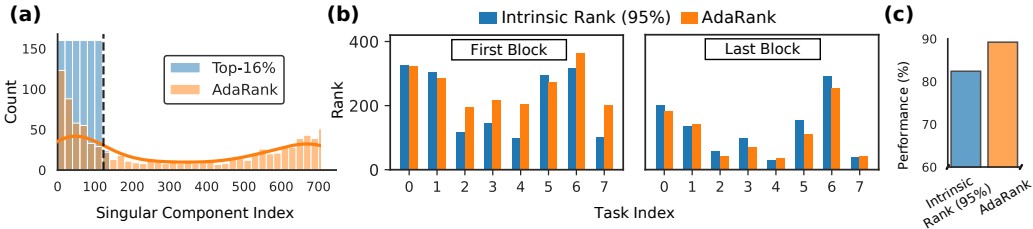

Figure 4: **(a)** Count of singular component indices selected by AdaRank, cumulated along 8 tasks. The black dashed line denotes the top-16% limit. **(b)** Comparison of ranks obtained from final masks after applying AdaRank, against the intrinsic rank for the MLP layers in the first (left) and last (right) blocks of ViT-B/32. **(c)** Performance comparison between merged model with top-$k$ truncation based on intrinsic rank and AdaRank (y-axis clipped for better visualization).

ponents. To quantify the performance gain from selecting the bottom components, we conduct an ablation study on the selection range. Starting from the best-performing top-16% truncation, we compare AdaRank applied only on top-$k$ range (i.e, $B_{ir} = 0$ always for $r > k$) with that used on the full range. As shown in Table 5, this variant also presents a significant performance gain, showing that removing conflicting top components is critical for mitigating task interference, as discussed in Section 3. However, AdaRank with a full range gains further by incorporating components from the indices outside the 16% range. We conjecture that bottom components provide valuable gains for their own task while introducing less inter-task interference than top components, aligning with the observations that they often correlate with fine details (Dabov et al., 2007; Candes & Plan, 2010; Lee et al., 2025).

**Effectiveness of Adaptive Rank Pruning.**
Our method is predicated on the idea that a better set of singular components that minimizes inter-task interference exists beyond the top-$k$ heuristic. To validate this, we first construct a supervised oracle with access to ground-truth labels of test data and optimize $B$ to minimize the multi-task cross-entropy loss, which serves as a direct metric for task interference (see Section 3). Figure 5 plots the cross-entropy loss of the oracle over optimization steps, where it shows that the oracle (blue curve) rapidly discovers a set of singular components that reduces the loss to the level identical to the individual models (black dashed line). This affirms that a far superior set of singular components exists beyond the naive top-$k$ selection

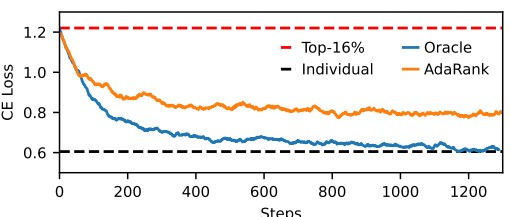

Figure 5: Cross-entropy loss during the optimization of $B$ initialized from top-16% truncation (black line). We compare optimizing AdaRank with cross-entropy loss to directly minimize the multi-task loss (blue curve) and entropy as surrogate loss (orange curve).

(red dashed line), and such a selection is sufficient to recover individual model performance. Next, we evaluate AdaRank, which leverages entropy as an unsupervised surrogate, starting from the same top-16% truncation. AdaRank (orange curve) also finds a substantially better set of components than the top-$k$ baseline, which validates that entropy minimization over $B$ effectively identifies beneficial singular components even without supervision.

**Adaptive Rank Assignment Across Tasks and Layers.** We now examine the ranks learned by our method, defined as the number of preserved singular components (i.e., where $B_{ir} = 1$). In Figure 4 **(b)**, we compare these learned ranks with the intrinsic ranks of task vectors that were shown in Section 3 and Figure 2, which capture 95% of the total spectral energy. We find that AdaRank's learned ranks are closely correlated with the intrinsic ranks across tasks and layers, suggesting that AdaRank automatically adapts to each task vector's complexity. Furthermore, despite preserving a similar rank, AdaRank outperforms top-$k$ truncation based on intrinsic rank (Figure 4 **(c)**), because the latter's approach still includes detrimental top components. Therefore, our method's superior performance stems from its ability to not only assign task- and layer-specific ranks but also to com-

Table 5: Ablation on the range of singular component selection with merging ViT-B/32 fine-tuned on 8 tasks.

| Method | TA | CART |
|---|---|---|
| Top-16% Fixed | 68.8 | 84.7 |
| AdaRank (Top-16%) | 87.5 | 88.5 |
| AdaRank (Full) | **87.9** | **89.2** |

Table 6: Ablation on learnable components with merging ViT-B/32 fine-tuned on 8 tasks. $\lambda$ denotes the coefficient, while $B$ is the binary mask.

| Component | | Method | | |
|---|---|---|---|---|
| $\lambda$ | $B$ | TA | TSV-M | CART |
| $\times$ | $\times$ | 69.1 | 83.6 | 84.7 |
| $\checkmark$ | $\times$ | 80.1 | 87.1 | 85.9 |
| $\times$ | $\checkmark$ | 79.9 | 87.2 | 88.7 |
| $\checkmark$ | $\checkmark$ | **87.9** | **88.9** | **89.2** |

Table 7: Performance of AdaRank with varying amounts of test data on ViT-B/32. The leftmost columns report the baseline without TTA (0%) and AdaMerging using the full test set for reference.

| | Method | No TTA | AdaMerging | | **AdaRank** | | | |
|---|---|---|---|---|---|---|---|---|
| | Amount of Test Data | 0% | 100% | 1% | 5% | 10% | 50% | 100% |
| 8 tasks | TA | 69.2 | 80.1 | 81.2 | 84.2 | 85.2 | 87.2 | 87.9 |
| | CART | 65.4 | 76.7 | 76.8 | 78.2 | 79.0 | 81.3 | 82.1 |
| 14 tasks | TA | 61.0 | 69.2 | 70.0 | 71.3 | 73.0 | 76.7 | 81.4 |
| | CART | 84.7 | 85.9 | 86.1 | 87.1 | 87.7 | 88.6 | 89.2 |
| 20 tasks | TA | 79.5 | 82.3 | 82.8 | 84.4 | 85.3 | 85.8 | 86.2 |
| | CART | 76.8 | 82.7 | 82.7 | 84.3 | 85.2 | 85.8 | 86.4 |

pose them by pruning interfering top components and selectively incorporating valuable bottom ones. For additional discussion and visualization of learned masks, see Appendix C.2.

**Ablation on Learnable Components.** We conduct an ablation to isolate the effects of adapting the coefficient $\lambda$ and binary mask $B$. As shown in Table 6, adapting either $\lambda$ or $B$ independently provides significant performance gains across all baselines. Adapting only $B$ yields improvements comparable to adapting $\lambda$ (TA), or surpassing (TSV-M, CART). These results indicate that selecting beneficial singular components makes a notable contribution for improving model merging performance, regardless of finding an appropriate value for the coefficient $\lambda$. Nonetheless, since both approaches orthogonally enhance multi-task performance, we employ both for our best-performing model.

**Robustness to the Amount of Test Data.** Lastly, we examine the robustness of AdaRank with respect to the amount of test data available for TTA. Table 7 reports the performance when varying the proportion of accessible test data from 1% to 100%. Even with only 1% of the test set, AdaRank achieves gains of 12.0% and 3.3% on the 8- and 20-task benchmarks, respectively, surpassing AdaMerging adapted with the full test set. This trend holds consistently across both TA and CART baselines, demonstrating that AdaRank is robust to limited data availability and it offers an effective solution for real-world scenarios. For results with NLP tasks, see Table 10 in appendix.

## 7 CONCLUSION

SVD-based model merging suffers from suboptimal performance, mainly due to inter-task interference induced by its heuristic top-$k$ approximation and its reliance on a fixed rank assignment across tasks and layers. To tackle this, we present AdaRank, a dynamic extension of SVD-based rank truncation for multi-task model merging, which uses test-time adaptation to learn a binary mask over singular components to guide the selection of beneficial singular components that minimize the interference. We empirically show that our method successfully prunes interfering components and assigns adaptive ranks that align with the intrinsic rank of each task and layer. The experimental results show that our method is a promising solution for enhancing multi-task performance on various scenarios of model merging, including merging models with various modalities and backbones.

**Ethics Statement.** We have carefully reviewed the ICLR Code of Ethics and confirm that this work adheres to its principles. The model checkpoints and datasets utilized in our work are publicly available and standard benchmarks in the field. To the best of our knowledge, our research does not introduce new ethical concerns.

**Reproducibility Statement.** We are committed to the reproducibility of our work. The Appendix provides comprehensive details of our method, including algorithm pseudo-code (Appendix A), implementation details and hyperparameters (Appendix B.1), and descriptions of all datasets involved (Appendix B.2). We have released a public code and model checkpoints of this work.

**LLM Usage.** We acknowledge the use of LLM in this work, strictly limited to refining and polishing the grammar of the manuscript for readability. All final contents are written and carefully verified by the authors.

## ACKNOWLEDGEMENTS

This work is supported by the National Research Foundation of Korea (RS-2024-00351212 and RS-2024-00436165), the Institute of Information & communications Tech- nology Planning & Evaluation (IITP) (RS-2024-00509279, RS-2022-II220926 , RS-2022-II220959, and RS-2019-II190075), funded by Korea Government (MSIT).

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

# A  ALGORITHM DETAILS

We summarize the full procedure of AdaRank in Algorithm 1, including the SVD preparation step for each baseline method and the test-time adaptation loop.

---
**Algorithm 1** AdaRank
---

// SVD Components Preparation
**Require:** Pretrained weight $\theta_0$, fine-tuned weights $\{\theta_i\}_{i=1}^T$, Method
**for** task $i = 1$ **to** $T$ **do**
  **if** Method is **TA** or **TSV-M** **then**
    $\tau_i \leftarrow \theta_i - \theta_0$
  **else if** Method is **CART** **then**
    $\bar{\theta} \leftarrow \frac{1}{T} \sum_{j=1}^T \theta_j$
    $\tau_i \leftarrow \theta_i - \bar{\theta}$
  **end if**
**end for**
**for** layer $l = 1$ **to** $L$ **do**
  **for** task $i = 1$ **to** $T$ **do**
    $U_i^l, \Sigma_i^l, V_i^l \leftarrow \text{SVD}(\tau_i^l)$
  **end for**
  **if** Method is **TSV-M** **then**
    $U^l \leftarrow [U_1^l | \ldots | U_T^l], \quad V^l \leftarrow [V_1^l | \ldots | V_T^l]$
    $U_\perp^l \leftarrow \text{whitening}(U^l), \quad V_\perp^l \leftarrow \text{whitening}(V^l)$
    Split $U_\perp^l, V_\perp^l$ back into per-task matrices $\{U_i^l\}_{i=1}^T, \{V_i^l\}_{i=1}^T$
  **end if**
**end for**

---

// Test-Time Adaptation
**Require:** Test datasets $\{\mathcal{D}_i\}_{i=1}^T$, task coefficients $\{\lambda^l\}$, learning rate $\eta$
**Initialize:** Mask parameters $B = \{B_i^l\}_{i=1,l=1}^{T,L}$
**repeat**
  Construct merged model $\theta(B)$ via Equation 5
  $\mathcal{L} \leftarrow \sum_{i=1}^T \mathbb{E}_{x_i \sim \mathcal{D}_i}[H(f(\theta(B), x_i))]$
  Update $B \leftarrow B - \eta \nabla_B \mathcal{L}$ using Straight-Through Estimator (Appendix A)
**until** converged
Construct final merged model $\theta^*$ using finalized $B^*$ via Equation 5
**Return** $\theta^*$

---

**Straight-Through Estimator.** We add a detailed explanation of how the Straight-Through Estimator (Bengio et al., 2013) works with our binary masks. In the case of a single mask parameter from $l$th layer of $i$th task vector $b_i^l \in \{0, 1\}$, we maintain a corresponding continuous parameter $\tilde{b}_i^l \in \mathbb{R}$. During the forward pass, we first apply the sigmoid function to constrain $\tilde{b}_i^l$ between 0 and 1, and then round the result to obtain a binary mask using a threshold of 0.5. Specifically, we compute $b_i^l = \mathbf{1}\{\sigma\left(\tilde{b}_i^l/T\right) \geq 0.5\}$, where $\sigma$ denotes sigmoid function, and $T$ is the temperature. During the backward pass, we pass the gradient through the continuous value of $\tilde{b}_i^l$. We constantly applied $T = 10$ in our experiment, which performed the best among $T \in \{1, 2, 5, 10\}$.

# B  EXPERIMENTAL SETUP

In this section, we explain our experiment settings, including the selection of hyperparameters and descriptions of the utilized datasets.

## B.1  IMPLEMENTATION DETAILS

**Initialization of Learnable Components.** As described in Section 6.1, we initialized the task-wise coefficient $\lambda$ and the binary mask $B$ based on the best-performing models for each method (Il-

harco et al., 2023; Choi et al., 2024; Gargiulo et al., 2025). For Task Arithmetic, we used fixed values of $\lambda = 0.3$ and $B = 1$ across all indices, consistent with the reported best-performing configuration. For CART, we performed a grid search over $\lambda \in \{0.1, 0.2, \dots, 3.0\}$ and selected $\lambda = 2.3$, which produced the highest performance. For $B$, we initialized $B = 1$ for the top 8%, 12%, 16%, and 32% of indices and chose the value of 16% that performed best. For TSV-M, we also searched for the best $\lambda$ with a grid search and used $\lambda = 1.0$. For $B$, we initialized the first $r = \frac{T}{d}$ entries as 1, where $T$ is the number of tasks, and $d$ is the number of total singular value entries, which is the best-performing setting denoted by the authors.

**Test-Time Adaptation.** For test-time adaptation (TTA), we share the same hyperparameters and TTA budget for all the adaptive merging and router-based merging baselines (Yang et al., 2023; Wei et al., 2025b; Lu et al., 2024; Tang et al., 2024b). We followed the settings from AdaMerging (Yang et al., 2023), using the Adam (Kingma & Ba, 2015) optimizer with a learning rate of $1 \times 10^{-3}$ for vision model merging and $5 \times 10^{-5}$ for language model merging. Both configurations used Adam betas of $(0.9, 0.999)$ and a test batch size of 16 per task.

**Computational Resources.** All merging, TTA, and evaluation experiments were conducted on a single NVIDIA RTX A6000 with 48GB of memory, except for the vision 8-task benchmark, which was performed on an NVIDIA RTX 3090 with 24GB of memory.

## B.2 DATASET DESCRIPTION

**Datasets for Vision Tasks.** In vision model merging experiments, we employed the codebases of Task Arithmetic (Ilharco et al., 2023) and AdaMerging (Yang et al., 2023). We used the same fine-tuned checkpoints as Task Arithmetic for the 8-task benchmark, while for the 14- and 20-task benchmarks, we utilized checkpoints provided by Consensus Merging (Wang et al., 2024).

For the 8-task benchmark of vision model merging experiments, we use the following datasets:

- **Stanford Cars (Cars)** (Krause et al., 2013) is a fine-grained car classification dataset containing 16,185 images across 196 classes.
- **DTD** (Cimpoi et al., 2014) is a texture classification benchmark comprising 5,640 images from 47 categories.
- **EuroSAT** (Helber et al., 2019) is a satellite image classification dataset of 27,000 labeled images distributed among 10 classes.
- **SVHN** (Netzer et al., 2011) is a real-world digit recognition dataset, split into 73,257 training and 26,032 test samples across 10 classes.
- **GTSRB** (Stallkamp et al., 2011) (German Traffic Sign Recognition Benchmark) is a traffic sign classification dataset with 39,209 training and 12,630 test images across 43 classes.
- **MNIST** (LeCun, 1998) is a foundational dataset of handwritten digits, containing 60,000 training and 10,000 test $28 \times 28$ grayscale images across 10 classes.
- **SUN397** (Xiao et al., 2016) is a large-scale scene understanding benchmark that includes 108,753 images from 397 distinct scene categories.
- **RESISC45** (Cheng et al., 2017) is a remote sensing image scene classification dataset, comprising 31,500 images across 45 scene classes (700 images per class).

For the 14-task benchmark, we extend the above set with six additional datasets:

- **CIFAR-100** (Krizhevsky, 2009) is a 100-class object recognition dataset with 60,000 $32 \times 32$ color images, split into 500 training and 100 test images per class.
- **STL-10** (Coates et al., 2011) is an image recognition benchmark containing 5,000 labeled training images and 8,000 test images across 10 classes, derived from a larger pool of unlabeled images.
- **Flowers102** (Nilsback & Zisserman, 2008) is a fine-grained flower classification dataset with 102 categories, where each class contains between 40 and 258 images.
- **Oxford-IIIT Pet** (Parkhi et al., 2012) is a fine-grained pet image classification dataset with 37 categories, containing approximately 200 images per class.

- **PCAM** (Veeling et al., 2018) is a medical imaging dataset for metastasis detection. It consists of 327,680 color images of histopathologic scans, each with a binary label.
- **FER2013** (Goodfellow et al., 2013) is a facial expression classification dataset with 7 expression classes, split into 28,709 training and 3,589 test examples.

For the 20-task benchmark, we further include six more datasets:

- **EMNIST** (Cohen et al., 2017) is a handwritten character recognition dataset, comprising 131,600 images distributed across 47 classes.
- **CIFAR-10** (Krizhevsky, 2009) is a canonical object recognition benchmark consisting of 60,000 $32 \times 32$ color images in 10 classes.
- **Food101** (Bossard et al., 2014) is a large-scale food image classification dataset containing 101,000 images across 101 distinct food categories.
- **FashionMNIST** (Xiao et al., 2017) is a dataset of fashion product images, designed as a drop-in replacement for MNIST. It contains 70,000 $28 \times 28$ grayscale images across 10 classes.
- **RenderedSST2** (Socher et al., 2013) is a dataset for evaluating OCR systems. It contains images of rendered sentences from the SST-2 dataset, split into 6,920 training and 1,821 test samples for a binary classification task.
- **KMNIST** (Clanuwat et al., 2018) is a dataset of handwritten Japanese *Kuzushiji* characters, containing 70,000 $28 \times 28$ grayscale images across 10 classes.

### B.3 Datasets for Language Tasks

For language model experiments, we adapt the codebase from EMR-Merging (Huang et al., 2024) and utilize checkpoints from DARE (Yu et al., 2024). We evaluate on seven text classification tasks from the widely-used GLUE benchmark (Wang et al., 2018).

- **CoLA** (Warstadt et al., 2019) is a binary classification task to determine if a sentence is grammatically acceptable. It contains 10.7k samples, and performance is measured by the Matthews Correlation Coefficient (MCC).
- **SST-2** (Socher et al., 2013) is a binary sentiment classification task on movie reviews, comprising 67k training, 872 validation, and 1.8k test samples.
- **MRPC** (Dolan & Brockett, 2005) is a binary classification task to determine if two sentences are semantically equivalent. The standard GLUE split consists of 3.7k training and 1.7k test pairs.
- **QQP** (Quora, 2017) is a binary classification task to identify whether two questions are semantically equivalent. The dataset contains over 400k question pairs.
- **MNLI** (Williams et al., 2018) is a three-class textual entailment task (entailment, contradiction, neutral) on sentence pairs, with over 433k pairs across its splits.
- **QNLI** (Rajpurkar et al., 2016) is a binary classification task derived from SQuAD, determining if a context sentence contains the answer to a question. It contains over 100k question-context pairs.
- **RTE** (Dagan et al., 2006) is a small binary entailment classification dataset aggregated from several sources, containing approximately 2.5k training samples.

## C Additional Results and Analyses

### C.1 Extended Analysis on SVD of Task Vectors

**Loss analysis on different models.** First, we verify that the trend in loss change induced by singular component addition is consistent across different scenarios. In Figure 6, we replicate the loss analysis on RoBERTa (Liu et al., 2019) and GPT-2 (Radford et al., 2019), where we individually add a singular component of a single task vector to a model merged with full-rank task vectors from the other tasks. Regardless of the change in architecture, we observe a similar trend as in the ViT: top components strongly benefit their own task loss, but some of those components also increase the net multi-task loss. This confirms that our finding is a general phenomenon across diverse model merging scenarios, not limited to ViT with image classification tasks.

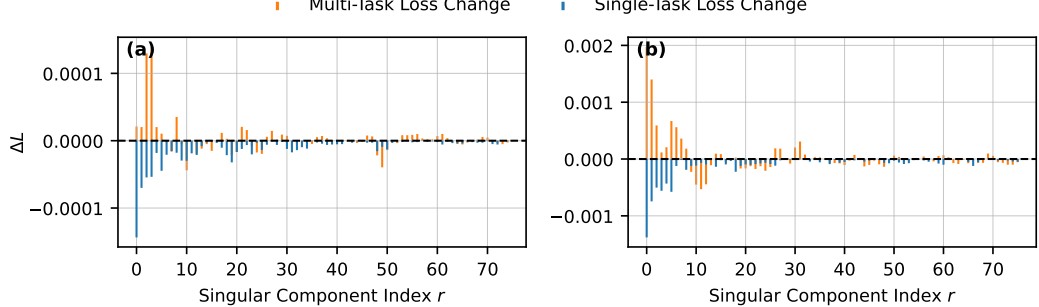

Figure 6: Net change in single-task and multi-task losses for **(a)** RoBERTa (Liu et al., 2019) and **(b)** GPT-2 (Radford et al., 2019), respectively, when each singular component of a target task vector is individually added to a model merged with full-rank vectors from other tasks. Task vectors are defined as deviation from pre-trained weight as Task Arithmetic Ilharco et al. (2023). Only Top-10% indices are presented for clarity.

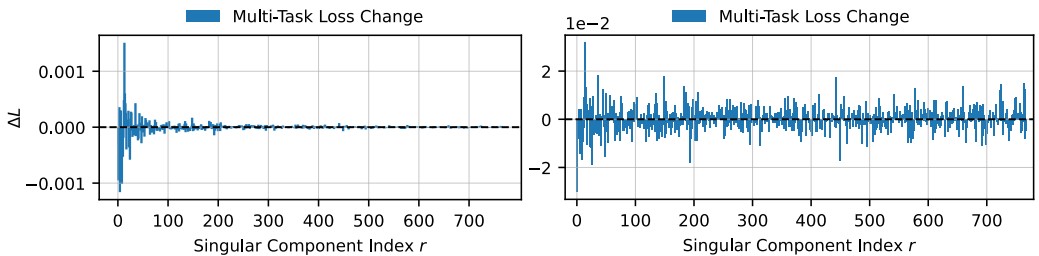

Figure 7: Net change in multi-task loss when each singular component of a target task vector is individually added to a model merged with full-rank vectors from other tasks. (Left) Loss change induced by adding scaled component $\mathbf{s}_i = \sigma_i \mathbf{u}_i \mathbf{v}_i^\top$. (Right) Loss change induced by adding component $\bar{\mathbf{s}}_i = \mathbf{u}_i \mathbf{v}_i^\top$.

**Condition for interference.** To understand why such components appear, we first analyze the change in multi-task loss $L(\theta) = \sum_{t=1}^{T} L_t(\theta)$ induced by adding a singular component. Consider adding an arbitrary singular component from task $i$, denoted as $\mathbf{s}_i = \sigma_i \mathbf{u}_i \mathbf{v}_i^\top$, to the model parameter $\theta$. We rewrite the loss change $\Delta L = L(\theta + \mathbf{s}_i) - L(\theta)$ in terms of the direction of the component $\bar{\mathbf{s}}_i = \mathbf{u}_i \mathbf{v}_i^\top$ and the singular value $\sigma_i$, using a second-order Taylor expansion:

$$\Delta L \approx \sigma_{ir} \nabla L^\top \bar{\mathbf{s}}_i + \frac{1}{2} \sigma_i^2 \bar{\mathbf{s}}_i^\top \mathbf{H} \bar{\mathbf{s}}_i, \tag{7}$$

where $\mathbf{H}$ is the Hessian of the loss at $\theta$. The value of $\Delta L$ is determined by two factors:

1. **Directional Alignment**: The alignment between the singular component direction $\bar{\mathbf{s}}_{ir}$ and the multi-task loss landscape (gradient and Hessian)

2. **Spectral Scale**: The magnitude of the singular value $\sigma_i$, which scales each first and second-order term.

We showed the value of $\Delta L$ as "Multi-Task Loss Change" plot in Figure 1, where it implicitly contains both the effects from scale and direction of the singular component.

**Singular Value Magnitude vs. Directional Alignment.** We now investigate the primary cause driving the interference in the top components. To decouple the effects of spectral scale and direction, we compare loss changes when adding (i) $\mathbf{s}_i = \sigma_i \mathbf{u}_i \mathbf{v}_i^\top$ (Figure 7 left) and (ii) $\bar{\mathbf{s}}_i = \mathbf{u}_i \mathbf{v}_i^\top$ (Figure 7 right), where singular values are removed. When we exclude the effect of the magnitude,

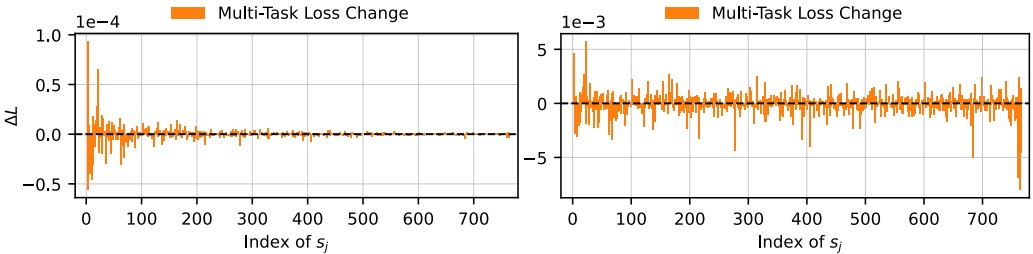

Figure 8: Change in multi-task loss caused by **joint interaction** of adding two singular components $\mathbf{s}_i$ and $\mathbf{s}_j$ together. We fix $\mathbf{s}_i$ as the top-1 component of task $i$ and sweep through the full indices of task $j$. (Left) Joint interaction measured with normalized components. (Right) Joint interaction measured with original scaled components.

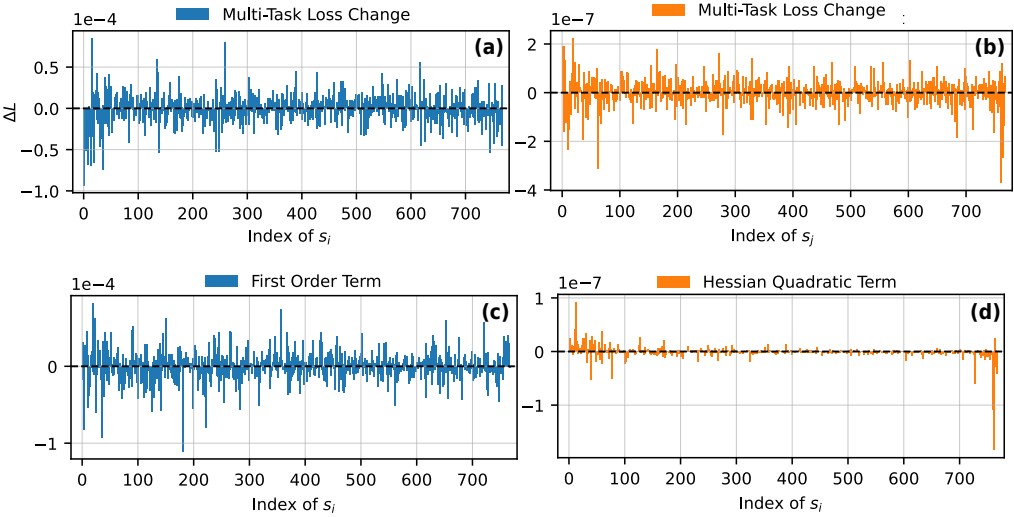

Figure 9: **(a)** Loss change induced by adding component $\tilde{\mathbf{s}}_i = \sigma_i \mathbf{u}_i \mathbf{v}_i^\top$. **(b)** $\Delta L_{\text{joint}}$ from adding $\tilde{\mathbf{s}}_i = \sigma_i \mathbf{u}_i \mathbf{v}_i^\top$ and $\tilde{\mathbf{s}}_j = \sigma_j \mathbf{u}_j \mathbf{v}_j^\top$ simultaneously. **(c), (d)** Values of the each terms in the Taylor expansion of the loss along singular directions (Equation 7): first-order term $\sigma_i \nabla L^\top \mathbf{s}_i$ **(c)** and second-order term $\sigma_i^2 \mathbf{s}_i^\top H \mathbf{s}_i$ **(d)**, respectively.

the distribution of the interfering components becomes nearly uniform along the spectral indices, indicating that direction alone does not strongly distinguish the interference caused by top and bottom components. However, restoring singular values amplifies the change induced by top components due to their large $\sigma$: while beneficial ones now help more, misaligned ones cause larger increases in loss.

**Alignment with Loss Curvature.** To further understand how singular directions interact with the loss landscape, we examine their alignment with the Hessian of the multi-task loss. Due to the heavy cost for calculating the full Hessian, we instead evaluate the quadratic term $\sigma_i^2 \bar{\mathbf{s}}_i^\top \mathbf{H} \bar{\mathbf{s}}_i$ from Equation 7 using Hessian-vector products (HVP). Although this does not give explicit principal components of $\mathbf{H}$, it captures the aggregate alignment of each singular direction with high-curvature regions of the loss. Figure 9 **(d)** plots this quadratic term. Interestingly, we observe a distinct pattern compared to the total loss change: top singular components tend to exhibit positive curvature $(\bar{\mathbf{s}}_i^\top \mathbf{H} \bar{\mathbf{s}}_i > 0)$, while many bottom components show negative values. This suggests that the Hessian does introduce a directional effect—top components are more aligned with upward curvature than bottom components.

However, this curvature effect is not the primary driver of interference in the top range. In Figure 9 **(c)**, we plot the first-order term $\bar{\sigma} \nabla L^\top \bar{\mathbf{s}}_i$. The first-order term exhibits behavior consistent with the actual loss change in Figure 9 **(a)**: interfering directions $(\nabla L^\top \bar{\mathbf{s}}_i > 0)$ appear across the

spectrum rather than being concentrated at the top indices. Taken together, these results indicate that while the Hessian alignment introduces a mild effect—top components tend to lie in regions of upward curvature—the first-order term dominates the overall loss change, and its interfering directions are distributed broadly across indices. Therefore, when comparing spectral scale and directional effects, the magnitude of the singular values remains the more dominant factor in shaping the top-range interference pattern observed in Section 3.

**Joint Interaction Effects.** Finally, we consider the change emerging from the joint addition of singular components. Consider simultaneously adding two singular components $\mathbf{s}_i$ and $\mathbf{s}_j$, each from task $i$ and $j$, to $\theta$. Using a second-order Taylor expansion, the multi-task loss change $\Delta L = L(\theta + \mathbf{s}_i + \mathbf{s}_j) - L(\theta)$ can be expressed as:

$$\Delta L \approx \underbrace{\sum_{p \in \{i,j\}} \left( \sigma_p \nabla L^\top \bar{\mathbf{s}}_p + \frac{1}{2}\sigma_p^2 \bar{\mathbf{s}}_p^\top \mathbf{H} \bar{\mathbf{s}}_p \right)}_{\text{Individual Terms}} + \underbrace{\sigma_i \sigma_j (\bar{\mathbf{s}}_i^\top \mathbf{H} \bar{\mathbf{s}}_j)}_{\text{Joint Term}}. \tag{8}$$

The first two terms correspond to the loss changes from adding each component individually, as in Equation 7, and are independent of the other component. The joint interaction arises from the last term, which is a Hessian inner product between two directions. As in the single-component case, this interaction is governed by both the directional alignment of two singular vectors with respect to the loss curvature and the size of the singular values. Empirically, we quantify this joint interaction by measuring:

$$\Delta L_{\text{joint}} = L(\theta + \mathbf{s}_i + \mathbf{s}_j) - (L(\theta + \mathbf{s}_i) + L(\theta + \mathbf{s}_j) - L(\theta)), \tag{9}$$

which isolates the additional loss change that occurs only when $\mathbf{s}_i$ and $\mathbf{s}_j$ are added together.

We fix $\mathbf{s}_i$ as the top-1 component from task $i$, and sweep over all singular components of task $j$ to measure this interaction. We observe a similar trend as in the single-component analysis (Figure 8); the directional interaction does not show a strong dependence on the index of $\mathbf{s}_j$, but interactions involving top components become much larger after accounting for their singular value. The joint effect is most amplified when both $\mathbf{s}_i$ and $\mathbf{s}_j$ originate from the top range, where their large singular values quadratically scale the interaction term. Therefore, joint interaction reinforces the same conclusion as the single-component addition: the risk of interference arises from misaligned directions being amplified by large singular values, especially for those prominent in the top range.

**Summary.** Overall, we decomposed the loss change into first-order ($\nabla L^\top \bar{\mathbf{s}}_i$), quadratic ($\bar{\mathbf{s}}_i^\top \mathbf{H} \bar{\mathbf{s}}_i$), and joint ($\bar{\mathbf{s}}_i^\top \mathbf{H} \bar{\mathbf{s}}_j$) terms. We found that only the quadratic term shows a mild concentration of interfering components in the top range, while the first-order and joint terms produce interfering directions broadly across the spectrum. This suggests that the Hessian does introduce a directional effect, but this effect is secondary to the dominant impact of singular value scaling. In other words, fixed top-$k$ selection is suboptimal primarily because large singular values amplify any misaligned direction in the range, and this amplification effect is the main driver of top-range interference. This analysis also explains why AdaRank selects directions beyond top-$k$ (Figure 4 **(a)**): since bottom components often exhibit lower or negative curvature, they can provide useful updates with reduced interference. Consistent with this, Table 5 confirms that AdaRank's primary gains come from pruning misaligned top components, while selecting non-top components yields smaller additional improvements.

## C.2 Mask Visualization.

To provide a visual illustration of our method's behavior, Figure 10 displays the final binary masks learned by AdaRank. We merged ViT-B/32 models fine-tuned with 8 tasks, which were initialized from the best-performing top-16% truncation. We observe two key patterns from these masks, which support the analysis in Section 6.3. First, the masks confirm that AdaRank prunes a significant portion of the initial top-$k$ singular components, while concurrently selecting numerous components from outside this range. Second, the masks exhibit apparent heterogeneity across both tasks and layers. When we compare the first block (top row) masks with those from the last block (bottom row), early layers preserve a broader range of indices and exhibit more uniform pruning patterns across tasks. In contrast, deeper layers display greater variability in the preserved indices, reflecting

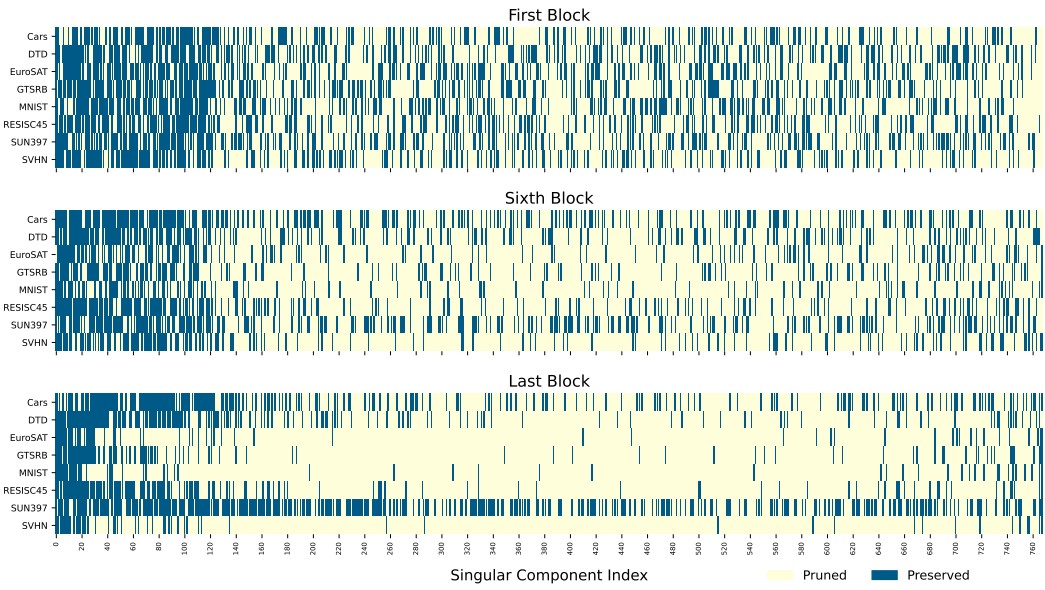

Figure 10: Binary masks derived from AdaRank for merging 8 fine-tuned ViT models. Masks corresponding to MLP Layers from the top, middle, and bottom blocks are plotted. The X-axis denotes the singular component index, ranging from 0 to 768, where each row in the Y-axis denotes individual tasks involved in merging. Each cell from the heatmap indicates whether the corresponding singular component from the task is preserved (blue) or pruned (yellow).

Table 8: Average multi-task performance on 8-task with merged ViT-B/32 and ViT-L/14 when applying Representation Surgery (Yang et al., 2024a) to AdaRank.

| Method | ViT-B/32 | | ViT-L/14 | |
|---|---|---|---|---|
| | AdaRank | AdaRank + Surgery | AdaRank | AdaRank + Surgery |
| TA | 87.9 | 88.8 (**+0.9**) | 93.0 | 93.3 (**+0.3**) |
| CART | 89.2 | 89.7 (**+0.5**) | 93.5 | 93.6 (**+0.1**) |
| TSV-M | 88.9 | 89.4 (**+0.5**) | 93.7 | 93.8 (**+0.1**) |

task-specific needs. Collectively, this visualization provides qualitative evidence for how AdaRank works: It adaptively builds the low-rank subspace for each task vector by pruning detrimental top singular components and tailoring the rank structure to the specific demands of each task and layer.

### C.3 ADDITIONAL BASELINES

In this section, we discuss the comparative study of additional merging baselines, including the compatibility of AdaRank with the post-calibration method (Yang et al., 2024a) and performance comparison to compression-based methods (Huang et al., 2024; Wang et al., 2024; Gargiulo et al., 2025).

**Post-Calibration Methods.** Representation Surgery (Yang et al., 2024a) and its following works (Yang et al., 2024b; Wei et al., 2025a) try to further enhance the merged model's performance by aligning the representations between the merged and individual models via a lightweight MLP adapted at test time. In Table 8, we report the performance of applying Representation Surgery over the best-performing AdaRank model, to show the compatibility of the post-merging method and AdaRank. Applying Representation Surgery consistently offers additional gains along all the baselines. Since Representation Surgery directly aligns the merged model's outputs to those of the non-merged experts, it complements AdaRank by reducing the residual gap left by entropy-only adaptation.

Table 9: Average multi-task performance on 8-task with merged ViT-B/32, ViT-L/14 compared to router and compression-based methods.

| Method | ViT-B/32 | ViT-L/14 | Task Index | Params. |
|---|---|---|---|---|
| Individual | 90.5 | 94.2 | - | 8× |
| TA+**AdaRank** | 87.9 | 93.0 | - | 1× |
| CART+**AdaRank** | **89.2** | 93.5 | - | 1× |
| TSV-M+**AdaRank** | 88.9 | **93.7** | - | 1× |
| EMR-Merging (Huang et al., 2024) | 88.8 | 93.5 | Required | 4.4× |
| TSV-C (Gargiulo et al., 2025) | 89.0 | 93.8 | Required | 2.5× |
| TALL-Masks (Wang et al., 2024) | **90.8** | **94.3** | Required | 2.2× |

Table 10: Performance of AdaRank on different amounts of test data, on merging RoBERTa / GPT-2 fine-tuned on 7 different tasks.

| Model | Method | No TTA | AdaMerging | AdaRank | | | | |
|---|---|---|---|---|---|---|---|---|
| | Amount of Test Data | 0% | 100% | 1% | 5% | 10% | 50% | 100% |
| RoBERTa | TA | 0.6718 | 0.6762 | 0.6774 | 0.6909 | 0.6937 | 0.6947 | 0.7032 |
| | CART | 0.6997 | 0.6954 | 0.7197 | 0.7087 | 0.7497 | 0.7453 | 0.7547 |
| GPT-2 | TA | 0.5910 | 0.5994 | 0.6330 | 0.6270 | 0.6304 | 0.6337 | 0.6328 |
| | CART | 0.6185 | 0.6207 | 0.6524 | 0.6484 | 0.6528 | 0.6511 | 0.6587 |

**Compression-Based Merging Methods.** Compression-based model merging methods preserve all task-specific parameters, similar to router-based methods, and assume the task identity is provided at inference to select the appropriate ones. This can be viewed as having an oracle router that always outputs the correct task index for each input. Therefore, these baselines naturally show higher merging performance since they do not collapse all parameters into a single parameter set, avoiding inter-task interference. However, their model size scale with the number of tasks, as do router-based methods. We consider the following baselines:

- **EMR-Merging** (Huang et al., 2024) involves three steps: Elect, Mask, and Rescale-Merging. It constructs and stores task-specific modulators during merging, applying them based on the task input index.
- **TSV-C** (Gargiulo et al., 2025)is a variant of TSV-M, which stores each low-rank task vector obtained from TSV-M and adds to the shared part with indexing.
- **TALL-Masks** (Wang et al., 2024) is a variant of Consensus Merging, which stores task-specific binary masks and applies them based on the task index rather than performing direct merging.

We compare results on merging 8 fine-tuned ViT-B/32 and ViT-L/14 models in Table 9. Despite a smaller size and lack of knowledge of the task index, AdaRank performs comparably, even outperforming some, emphasizing its effectiveness.

## C.4    FULL EXPERIMENT RESULTS

We present comprehensive experimental results, including individual task performances and additional baselines, for both vision models and language models. Full performance tables are provided from Table 11 to Table 18.

Table 11: Multi-Task performance comparison on 8 Vision Tasks with Merged ViT-B/32.

| Method | SUN397 | Cars | RESISC45 | EuroSAT | SVHN | GTSRB | MNIST | DTD | Avg. |
|---|---|---|---|---|---|---|---|---|---|
| Pretrained | 62.3 | 59.7 | 60.7 | 45.5 | 31.4 | 32.6 | 48.5 | 43.8 | 48.1 |
| Individual | 75.3 | 77.7 | 96.1 | 99.7 | 97.5 | 98.7 | 99.7 | 79.4 | 90.5 |
| Traditional MTL | 73.9 | 74.4 | 93.9 | 98.2 | 95.8 | 98.9 | 99.5 | 77.9 | 89.1 |
| Weight Averaging | 65.2 | 63.4 | 71.5 | 71.9 | 64.2 | 52.8 | 87.5 | 50.7 | 65.9 |
| Fisher Merging | 68.6 | 69.2 | 70.7 | 66.4 | 72.9 | 51.1 | 87.9 | 59.9 | 68.3 |
| RegMean | 65.3 | 63.5 | 75.6 | 78.6 | 78.1 | 67.4 | 93.7 | 52.0 | 71.8 |
| Task Arithmetic | 55.2 | 54.9 | 66.7 | 78.9 | 80.2 | 69.7 | 97.3 | 50.4 | 69.2 |
| Ties-Merging | 59.8 | 58.6 | 70.7 | 79.7 | 86.2 | 72.1 | 98.3 | 54.2 | 72.5 |
| Consensus-Ties | 62.5 | 61.8 | 76.3 | 81.6 | 82.0 | 80.5 | 97.3 | 56.0 | 74.8 |
| Consensus-TA | 63.9 | 62.2 | 76.1 | 84.2 | 84.2 | 76.6 | 97.4 | 57.5 | 75.2 |
| TSV-M | 67.2 | 70.8 | 86.3 | 94.6 | 91.0 | 92.3 | 99.3 | 68.9 | 83.8 |
| CART | 68.5 | 73.0 | 88.3 | 95.8 | 87.8 | 93.4 | 99.1 | 72.1 | 84.7 |
| AdaMerging | 64.5 | 68.1 | 79.2 | 93.8 | 87.0 | 91.9 | 97.5 | 59.1 | 80.1 |
| AdaMerging++ | 66.6 | 68.3 | 82.2 | 94.2 | 89.6 | 89.0 | 98.3 | 60.6 | 81.1 |
| TA+**AdaRank** | 71.1 | 79.1 | 91.3 | 97.2 | 94.2 | 98.3 | 99.2 | 72.7 | **87.9** |
| TSV-M+AdaMerging | 69.8 | 76.0 | 90.7 | 95.2 | 94.0 | 97.1 | 99.0 | 74.9 | 87.1 |
| TSV-M+**AdaRank** | 72.1 | 79.1 | 92.8 | 97.7 | 95.6 | 98.4 | 99.4 | 76.1 | **88.9** |
| CART+AdaMerging | 69.5 | 75.1 | 89.3 | 95.7 | 93.0 | 96.8 | 98.9 | 68.4 | 85.8 |
| CART+**AdaRank** | 72.1 | 78.9 | 93.3 | 98.4 | 95.6 | 98.8 | 99.4 | 76.9 | **89.2** |

Table 12: Multi-Task performance comparison on 8 Vision Tasks with Merged ViT-L/14.

| Method | SUN397 | Cars | RESISC45 | EuroSAT | SVHN | GTSRB | MNIST | DTD | Avg. |
|---|---|---|---|---|---|---|---|---|---|
| Pretrained | 68.3 | 77.8 | 71.0 | 62.4 | 58.4 | 50.6 | 76.4 | 55.3 | 65.0 |
| Individual | 82.3 | 92.4 | 97.4 | 99.9 | 98.1 | 99.2 | 99.7 | 84.1 | 94.1 |
| Traditional MTL | 80.8 | 90.6 | 96.3 | 96.3 | 97.6 | 99.1 | 99.6 | 84.4 | 93.1 |
| Weight Averaging | 72.1 | 81.6 | 82.6 | 91.9 | 78.2 | 70.7 | 97.1 | 62.8 | 79.6 |
| Fisher Merging | 69.2 | 88.6 | 87.5 | 93.5 | 80.6 | 74.8 | 93.3 | 70.0 | 82.2 |
| RegMean | 73.3 | 81.8 | 86.1 | 97.0 | 88.0 | 84.2 | 98.5 | 60.8 | 83.7 |
| Task Arithmetic | 73.9 | 82.1 | 86.6 | 94.1 | 87.9 | 86.7 | 98.9 | 65.6 | 84.5 |
| Ties-Merging | 76.5 | 85.0 | 89.3 | 96.3 | 90.3 | 83.3 | 99.0 | 68.9 | 86.1 |
| Consensus-Ties | 74.9 | 83.6 | 88.7 | 96.6 | 90.5 | 93.2 | 99.1 | 71.1 | 87.2 |
| Consensus-TA | 74.5 | 82.2 | 88.8 | 94.2 | 92.6 | 93.3 | 99.2 | 67.8 | 86.6 |
| TSV-M | 78.0 | 90.0 | 93.4 | 99.0 | 94.8 | 96.3 | 99.5 | 78.8 | 91.2 |
| CART | 79.3 | 90.4 | 95.4 | 99.3 | 96.1 | 98.3 | 99.6 | 82.5 | 92.6 |
| AdaMerging | 79.0 | 90.3 | 90.8 | 96.2 | 93.4 | 98.0 | 99.0 | 79.9 | 90.8 |
| AdaMerging++ | 79.4 | 90.3 | 91.6 | 97.4 | 93.4 | 97.5 | 99.0 | 79.2 | 91.0 |
| TA+**AdaRank** | 80.4 | 92.4 | 94.5 | 98.8 | 96.6 | 99.1 | 99.4 | 82.3 | **92.9** |
| TSV-M+AdaMerging | 79.1 | 91.8 | 95.1 | 98.9 | 96.9 | 98.9 | 99.5 | 83.8 | 93.0 |
| TSV-M+**AdaRank** | 81.0 | 92.5 | 96.2 | 99.5 | 97.6 | 99.0 | 99.5 | 84.0 | **93.7** |
| CART+AdaMerging | 80.1 | 91.5 | 94.7 | 99.3 | 96.8 | 98.9 | 99.5 | 83.6 | 93.1 |
| CART+**AdaRank** | 80.6 | 92.1 | 96.0 | 99.7 | 97.0 | 98.8 | 99.4 | 83.8 | **93.4** |

Table 13: Multi-Task performance comparison on 14 Vision Tasks with Merged ViT-B/32.

| Method | SUN397 | Cars | RESISC45 | EuroSAT | SVHN | GTSRB | MNIST |
|---|---|---|---|---|---|---|---|
| Pretrained | 62.3 | 59.7 | 60.7 | 45.5 | 31.4 | 32.6 | 48.5 |
| Individual | 75.3 | 77.7 | 96.1 | 99.7 | 97.5 | 98.7 | 99.7 |
| Weight Averaging | 64.2 | 60.7 | 67.2 | 64.6 | 49.4 | 43.5 | 76.2 |
| Task Arithmetic | 63.9 | 59.5 | 67.5 | 67.7 | 52.9 | 47.0 | 80.8 |
| Ties-Merging | 65.1 | 61.8 | 68.3 | 63.7 | 51.3 | 45.9 | 80.0 |
| Consensus-Ties | 63.6 | 58.8 | 69.7 | 71.9 | 56.2 | 61.2 | 88.3 |
| Consensus-TA | 62.8 | 54.8 | 68.5 | 76.0 | 69.3 | 63.0 | 93.5 |
| TSV-M | 66.3 | 62.1 | 81.2 | 91.7 | 82.4 | 83.6 | 98.8 |
| CART | 68.3 | 60.6 | 86.1 | 91.3 | 72.7 | 82.6 | 98.1 |
| AdaMerging | 64.3 | 68.5 | 81.7 | 92.6 | 86.6 | 90.8 | 97.5 |
| TA+**AdaRank** | 69.2 | 77.3 | 91.3 | 95.9 | 94.1 | 97.1 | 99.1 |
| TSV-M+AdaMerging | 69.3 | 73.6 | 88.8 | 95.3 | 93.8 | 95.0 | 98.5 |
| TSV-M+**AdaRank** | 70.4 | 76.2 | 91.8 | 98.4 | 95.3 | 97.1 | 99.2 |
| CART+AdaMerging | 67.4 | 72.5 | 87.8 | 96.0 | 90.9 | 95.6 | 98.6 |
| CART+**AdaRank** | 70.7 | 77.0 | 91.1 | 98.7 | 94.4 | 97.8 | 99.3 |

| Method | DTD | CIFAR100 | FER2013 | Flowers | Pet | PCAM | STL10 |
|---|---|---|---|---|---|---|---|
| Pretrained | 43.8 | 64.2 | 39.0 | 66.3 | 87.4 | 60.6 | 97.1 |
| Individual | 79.4 | 89.3 | 73.0 | 90.5 | 91.1 | 87.9 | 98.0 |
| Weight Averaging | 47.2 | 69.8 | 41.6 | 68.2 | 88.1 | 61.9 | 97.2 |
| Task Arithmetic | 48.2 | 69.6 | 42.9 | 67.6 | 87.5 | 63.2 | 96.7 |
| Ties-Merging | 48.7 | 69.7 | 42.4 | 68.1 | 88.0 | 62.1 | 97.2 |
| Consensus-Ties | 51.8 | 67.9 | 95.4 | 65.7 | 86.2 | 72.3 | 45.3 |
| Consensus-TA | 52.4 | 66.6 | 45.3 | 68.3 | 86.9 | 77.0 | 95.6 |
| TSV-M | 64.6 | 72.0 | 62.3 | 75.3 | 90.4 | 84.5 | 97.2 |
| CART | 67.7 | 75.6 | 60.9 | 81.6 | 90.2 | 79.7 | 97.6 |
| AdaMerging | 60.2 | 67.3 | 53.1 | 73.8 | 87.9 | 53.8 | 96.3 |
| TA+**AdaRank** | 72.5 | 75.6 | 45.4 | 82.1 | 92.2 | 59.5 | 97.5 |
| TSV-M+AdaMerging | 67.4 | 72.5 | 87.8 | 96.0 | 90.9 | 95.6 | 98.6 |
| TSV-M+**AdaRank** | 75.7 | 80.4 | 65.8 | 88.9 | 92.5 | 86.3 | 98.0 |
| CART+AdaMerging | 71.2 | 71.5 | 60.0 | 80.5 | 87.8 | 75.6 | 96.3 |
| CART+**AdaRank** | 75.6 | 79.4 | 61.4 | 89.1 | 91.7 | 83.0 | 97.7 |

Table 14: Multi-Task performance comparison on 14 Vision Tasks with Merged ViT-L/14.

| Method | SUN397 | Cars | RESISC45 | EuroSAT | SVHN | GTSRB | MNIST |
|---|---|---|---|---|---|---|---|
| Pretrained | 68.3 | 77.8 | 71.0 | 62.4 | 58.4 | 50.6 | 76.4 |
| Individual | 82.3 | 92.4 | 97.4 | 99.9 | 98.1 | 99.2 | 99.7 |
| Weight Averaging | 70.9 | 79.7 | 78.0 | 84.1 | 72.8 | 61.7 | 93.9 |
| Task Arithmetic | 72.1 | 76.4 | 81.1 | 89.3 | 81.7 | 75.4 | 97.9 |
| Ties-Merging | 74.0 | 78.8 | 83.7 | 90.7 | 83.0 | 70.7 | 98.1 |
| Consensus-Ties | 72.1 | 75.6 | 84.6 | 95.4 | 87.8 | 83.4 | 97.9 |
| Consensus-TA | 73.5 | 76.8 | 85.4 | 91.4 | 87.3 | 84.7 | 98.7 |
| TSV-M | 75.8 | 86.1 | 92.3 | 98.0 | 93.6 | 94.3 | 99.5 |
| CART | 77.9 | 86.0 | 94.1 | 98.8 | 92.8 | 95.9 | 99.5 |
| AdaMerging | 76.4 | 91.2 | 91.0 | 97.7 | 94.5 | 97.2 | 98.9 |
| TA+**AdaRank** | 78.7 | 92.6 | 94.8 | 98.4 | 95.7 | 98.5 | 99.0 |
| TSV-M+AdaMerging | 78.1 | 91.1 | 93.9 | 99.0 | 95.9 | 98.2 | 99.3 |
| TSV-M+**AdaRank** | 79.8 | 91.9 | 95.5 | 99.2 | 97.1 | 98.7 | 99.5 |
| CART+AdaMerging | 79.8 | 91.8 | 94.5 | 98.2 | 95.2 | 98.1 | 99.1 |
| CART+**AdaRank** | 79.7 | 92.0 | 95.0 | 98.8 | 96.5 | 98.6 | 99.3 |

| Method | DTD | CIFAR100 | FER2013 | Flowers | Pet | PCAM | STL10 |
|---|---|---|---|---|---|---|---|
| Pretrained | 55.3 | 75.8 | 38.2 | 79.1 | 93.6 | 51.2 | 99.4 |
| Individual | 84.1 | 93.3 | 77.0 | 97.9 | 95.5 | 90.3 | 99.5 |
| Weight Averaging | 59.7 | 82.7 | 42.5 | 80.5 | 94.7 | 74.2 | 99.4 |
| Task Arithmetic | 60.1 | 81.1 | 46.7 | 77.5 | 95.1 | 81.1 | 98.8 |
| Ties-Merging | 62.1 | 82.7 | 49.6 | 66.6 | 94.7 | 80.1 | 98.9 |
| Consensus-Ties | 65.5 | 80.4 | 47.5 | 76.7 | 94.4 | 80.7 | 98.5 |
| Consensus-TA | 62.9 | 80.7 | 51.4 | 76.9 | 95.3 | 82.6 | 98.6 |
| TSV-M | 74.5 | 85.6 | 69.0 | 87.9 | 96.1 | 83.9 | 99.5 |
| CART | 78.7 | 87.2 | 66.1 | 90.3 | 96.0 | 79.0 | 99.6 |
| AdaMerging | 79.5 | 84.3 | 49.5 | 95.1 | 95.5 | 82.4 | 99.1 |
| TA+**AdaRank** | 79.9 | 86.9 | 52.1 | 93.3 | 96.1 | 86.8 | 99.4 |
| TSV-M+AdaMerging | 82.9 | 87.9 | 74.1 | 96.3 | 96.0 | 85.0 | 99.4 |
| TSV-M+**AdaRank** | 82.6 | 89.1 | 74.3 | 97.9 | 96.4 | 88.8 | 99.5 |
| CART+AdaMerging | 82.0 | 86.8 | 75.2 | 94.5 | 96.5 | 74.7 | 99.4 |
| CART+**AdaRank** | 81.9 | 86.0 | 71.8 | 94.4 | 96.4 | 89.9 | 99.5 |

Table 15: Multi-Task performance comparison on 20 Vision Tasks with Merged ViT-B/32.

| Method | SUN397 | Cars | RESISC45 | EuroSAT | SVHN | GTSRB | MNIST | DTD | CIFAR100 | FER2013 |
|---|---|---|---|---|---|---|---|---|---|---|
| Pretrained | 62.3 | 59.7 | 60.7 | 45.5 | 31.4 | 32.6 | 48.5 | 43.8 | 64.2 | 39.0 |
| Individual | 75.3 | 77.7 | 96.1 | 99.7 | 97.5 | 98.7 | 99.7 | 79.4 | 89.3 | 73.0 |
| Weight Averaging | 59.6 | 46.0 | 56.3 | 41.3 | 70.0 | 64.6 | 64.0 | 69.3 | 96.9 | 66.5 |
| Task Arithmetic | 64.1 | 59.4 | 64.6 | 56.6 | 47.3 | 41.4 | 70.5 | 46.2 | 69.2 | 41.0 |
| Ties-Merging | 64.5 | 57.0 | 68.8 | 59.4 | 48.0 | 48.0 | 78.3 | 49.5 | 70.6 | 43.3 |
| Consensus-Ties | 64.4 | 58.9 | 67.2 | 54.4 | 51.1 | 47.9 | 77.5 | 48.4 | 67.6 | 96.3 |
| Consensus-TA | 63.6 | 52.5 | 65.3 | 64.1 | 63.1 | 52.6 | 88.0 | 49.1 | 65.6 | 42.0 |
| TSV-M | 64.3 | 52.0 | 75.9 | 87.1 | 75.2 | 76.8 | 94.6 | 61.1 | 68.1 | 58.2 |
| CART | 65.3 | 38.1 | 81.3 | 88.7 | 70.0 | 77.4 | 96.2 | 64.6 | 73.7 | 59.9 |
| AdaMerging | 62.1 | 66.3 | 78.7 | 92.1 | 72.7 | 90.6 | 93.6 | 57.6 | 66.3 | 48.4 |
| TA+**AdaRank** | 68.1 | 74.4 | 90.7 | 95.6 | 92.0 | 96.0 | 96.9 | 68.5 | 75.6 | 43.8 |
| TSV-M+AdaMerging | 67.3 | 70.2 | 86.2 | 93.9 | 89.6 | 92.4 | 95.9 | 71.4 | 73.1 | 66.7 |
| TSV-M+**AdaRank** | 68.8 | 74.4 | 92.0 | 98.0 | 94.6 | 95.7 | 97.6 | 76.0 | 77.9 | 63.4 |
| CART+AdaMerging | 67.3 | 71.2 | 86.3 | 96.6 | 88.3 | 95.0 | 96.4 | 71.2 | 72.4 | 54.4 |
| CART+**AdaRank** | 69.5 | 75.7 | 91.7 | 97.6 | 93.6 | 96.8 | 97.4 | 73.4 | 77.6 | 61.2 |

| Method | Flowers | Pet | PCAM | STL10 | EMNIST | CIFAR10 | Food101 | FMNIST | R-SST2 | KMNIST |
|---|---|---|---|---|---|---|---|---|---|---|
| Pretrained | 66.3 | 87.4 | 60.6 | 97.1 | 17.2 | 89.8 | 82.6 | 63.0 | 58.6 | 9.8 |
| Individual | 90.5 | 91.1 | 87.9 | 98.0 | 99.8 | 97.9 | 89.1 | 95.5 | 74.4 | 98.6 |
| Weight Averaging | 87.6 | 62.2 | 40.8 | 31.6 | 92.8 | 81.1 | 70.8 | 60.5 | 8.5 | 47.5 |
| Task Arithmetic | 66.7 | 87.7 | 62.4 | 96.9 | 32.9 | 92.7 | 81.1 | 70.7 | 60.4 | 8.7 |
| Ties-Merging | 71.6 | 85.3 | 64.4 | 96.0 | 39.9 | 93.5 | 75.9 | 72.7 | 64.7 | 12.4 |
| Consensus-Ties | 67.1 | 86.9 | 67.0 | 42.8 | 41.0 | 92.4 | 79.4 | 74.9 | 60.8 | 11.4 |
| Consensus-TA | 66.4 | 85.9 | 72.6 | 95.4 | 53.4 | 92.3 | 75.1 | 74.7 | 62.7 | 15.6 |
| TSV-M | 71.2 | 88.6 | 84.5 | 96.4 | 95.3 | 93.8 | 77.3 | 85.4 | 70.2 | 57.2 |
| CART | 77.6 | 87.7 | 74.8 | 97.0 | 93.1 | 94.8 | 77.1 | 86.5 | 68.6 | 63.4 |
| AdaMerging | 65.7 | 87.0 | 54.6 | 96.6 | 21.5 | 90.5 | 80.7 | 82.9 | 65.0 | 10.8 |
| TA+**AdaRank** | 75.5 | 91.9 | 60.7 | 97.3 | 95.9 | 94.6 | 83.8 | 91.0 | 69.3 | 66.3 |
| TSV-M+AdaMerging | 80.1 | 90.5 | 78.3 | 97.1 | 94.6 | 93.9 | 82.3 | 87.2 | 70.7 | 89.1 |
| TSV-M+**AdaRank** | 87.7 | 91.8 | 84.3 | 97.9 | 97.1 | 95.4 | 84.8 | 91.9 | 74.0 | 95.7 |
| CART+AdaMerging | 79.6 | 86.4 | 80.0 | 96.9 | 95.1 | 91.5 | 79.8 | 82.7 | 70.0 | 92.5 |
| CART+**AdaRank** | 85.6 | 91.7 | 83.8 | 97.5 | 96.6 | 94.8 | 83.6 | 91.0 | 73.5 | 96.0 |

Table 16: Multi-Task performance comparison on 20 Vision Tasks with Merged ViT-L/14.

| Method | SUN397 | Cars | RESISC45 | EuroSAT | SVHN | GTSRB | MNIST | DTD | CIFAR100 | FER2013 |
|---|---|---|---|---|---|---|---|---|---|---|
| Pretrained | 68.3 | 77.8 | 71.0 | 62.4 | 58.4 | 50.6 | 76.4 | 55.3 | 75.8 | 38.2 |
| Individual | 82.3 | 92.4 | 97.4 | 99.9 | 98.1 | 99.2 | 99.7 | 84.1 | 93.3 | 77.0 |
| Weight Averaging | 70.3 | 78.6 | 76.2 | 79.0 | 70.8 | 59.0 | 92.6 | 58.1 | 82.6 | 40.6 |
| Task Arithmetic | 71.3 | 76.6 | 77.8 | 82.9 | 75.6 | 65.4 | 95.8 | 59.3 | 81.8 | 41.9 |
| Ties-Merging | 72.5 | 75.4 | 79.3 | 82.6 | 78.8 | 64.7 | 96.6 | 60.2 | 80.7 | 44.8 |
| Consensus-TA | 72.6 | 76.2 | 82.4 | 86.9 | 82.1 | 76.7 | 97.4 | 61.6 | 80.5 | 45.4 |
| Consensus-Ties | 72.1 | 71.5 | 80.6 | 85.1 | 82.5 | 78.5 | 96.1 | 63.1 | 77.4 | 44.5 |
| TSV-M | 74.4 | 81.1 | 90.6 | 96.3 | 90.0 | 90.8 | 97.3 | 71.4 | 82.4 | 63.9 |
| CART | 76.3 | 75.3 | 92.4 | 97.9 | 89.9 | 94.1 | 98.5 | 76.1 | 84.8 | 62.5 |
| AdaMerging | 75.2 | 90.7 | 91.4 | 97.6 | 88.6 | 97.0 | 97.7 | 74.0 | 83.2 | 47.9 |
| TA+**AdaRank** | 77.3 | 91.7 | 94.7 | 97.4 | 93.2 | 98.1 | 98.0 | 75.9 | 85.0 | 54.4 |
| TSV-M+AdaMerging | 77.4 | 90.4 | 93.2 | 98.3 | 95.9 | 97.8 | 98.5 | 82.0 | 86.6 | 73.5 |
| TSV-M+**AdaRank** | 79.1 | 91.4 | 95.0 | 99.0 | 97.1 | 98.1 | 98.7 | 81.6 | 88.5 | 73.6 |
| CART+AdaMerging | 79.3 | 91.1 | 93.8 | 98.4 | 93.9 | 97.5 | 97.7 | 81.4 | 85.9 | 74.1 |
| CART+**AdaRank** | 79.9 | 91.5 | 94.7 | 98.6 | 94.8 | 98.2 | 97.4 | 80.4 | 86.5 | 70.7 |

| Method | Flowers | Pet | PCAM | STL10 | EMNIST | CIFAR10 | Food101 | FMNIST | R-SST2 | KMNIST |
|---|---|---|---|---|---|---|---|---|---|---|
| Pretrained | 79.1 | 93.6 | 51.2 | 99.4 | 15.6 | 95.6 | 92.3 | 66.9 | 68.9 | 10.4 |
| Individual | 97.9 | 95.5 | 90.3 | 99.5 | 99.8 | 99.2 | 95.5 | 95.8 | 85.4 | 98.8 |
| Weight Averaging | 80.0 | 94.5 | 71.0 | 99.4 | 36.3 | 97.3 | 92.5 | 76.3 | 67.4 | 11.5 |
| Task Arithmetic | 78.2 | 94.9 | 76.1 | 99.0 | 55.2 | 97.4 | 90.9 | 80.5 | 66.8 | 17.5 |
| Ties-Merging | 69.1 | 94.7 | 75.4 | 98.7 | 75.5 | 97.3 | 90.3 | 82.6 | 69.1 | 28.8 |
| Consensus-TA | 77.8 | 95.4 | 81.5 | 98.9 | 82.7 | 97.1 | 90.9 | 84.5 | 70.6 | 34.4 |
| Consensus-Ties | 74.8 | 94.6 | 78.9 | 98.3 | 79.7 | 96.3 | 87.5 | 81.6 | 65.9 | 43.9 |
| TSV-M | 85.6 | 95.9 | 85.0 | 99.3 | 99.3 | 97.9 | 92.3 | 91.0 | 82.9 | 77.7 |
| CART | 87.9 | 95.8 | 80.7 | 99.3 | 98.5 | 98.3 | 92.6 | 91.8 | 80.0 | 85.8 |
| AdaMerging | 95.1 | 95.4 | 50.3 | 99.1 | 96.3 | 97.2 | 92.7 | 89.7 | 82.5 | 94.3 |
| TA+**AdaRank** | 92.0 | 95.9 | 66.3 | 99.3 | 97.5 | 97.8 | 93.8 | 91.6 | 85.7 | 97.0 |
| TSV-M+AdaMerging | 97.8 | 96.2 | 91.4 | 99.6 | 98.0 | 98.1 | 94.1 | 92.7 | 84.1 | 96.6 |
| TSV-M+**AdaRank** | 97.8 | 96.2 | 91.4 | 99.6 | 98.7 | 98.5 | 94.7 | 93.2 | 85.3 | 97.6 |
| CART+AdaMerging | 95.7 | 96.5 | 79.2 | 99.4 | 98.1 | 97.8 | 93.4 | 91.5 | 85.4 | 96.7 |
| CART+**AdaRank** | 93.1 | 96.4 | 89.8 | 99.4 | 98.3 | 98.2 | 94.0 | 92.1 | 85.0 | 97.5 |

Table 17: Multi-task performance on 7 NLP tasks with merged RoBERTa model.

| Method | CoLA | SST2 | MRPC | QQP | MNLI | QNLI | RTE | Average |
|---|---|---|---|---|---|---|---|---|
| Individual | 0.6018 | 0.9404 | 0.8922 | 0.9141 | 0.872 | 0.9271 | 0.7906 | 0.8483 |
| Weight Averaging | 0.1808 | 0.8188 | 0.7794 | 0.7960 | 0.4383 | 0.7106 | 0.6173 | 0.6202 |
| Task Arithmetic (TA) | 0.2330 | 0.8658 | 0.7868 | 0.8395 | 0.6371 | 0.7304 | 0.6101 | 0.6718 |
| Ties-Merging | 0.2499 | 0.8349 | 0.7868 | 0.8515 | 0.6072 | 0.7580 | 0.4224 | 0.6444 |
| TSV-M | 0.3324 | 0.8716 | 0.8382 | 0.8598 | 0.5897 | 0.7274 | 0.4657 | 0.6693 |
| CART | 0.3092 | 0.9197 | 0.8088 | 0.7953 | 0.5767 | 0.7772 | 0.7112 | 0.6997 |
| AdaMerging | -0.0359 | 0.9266 | 0.7721 | 0.8221 | 0.7880 | 0.7961 | 0.6643 | 0.6762 |
| TA+**AdaRank** | 0.1401 | 0.9151 | 0.777 | 0.7963 | 0.7814 | 0.8409 | 0.6715 | **0.7032** |
| TSV-M+**AdaRank** | 0.3031 | 0.8761 | 0.8382 | 0.8519 | 0.7872 | 0.7922 | 0.6679 | **0.7309** |
| CART+**AdaRank** | 0.3710 | 0.9346 | 0.8309 | 0.8016 | 0.7698 | 0.8854 | 0.6895 | **0.7547** |

Table 18: Multi-task performance on 7 NLP tasks with merged GPT-2 model.

| Method | CoLA | SST2 | MRPC | QQP | MNLI | QNLI | RTE | Average |
|---|---|---|---|---|---|---|---|---|
| Individual | 0.4077 | 0.9118 | 0.8039 | 0.3964 | 0.8200 | 0.8827 | 0.6534 | 0.7680 |
| Weight Averaging | 0.1214 | 0.5252 | 0.5098 | 0.7670 | 0.5925 | 0.5761 | 0.4477 | 0.5057 |
| Task Arithmetic (TA) | -0.0019 | 0.8360 | 0.6961 | 0.8182 | 0.7188 | 0.7049 | 0.4729 | 0.6064 |
| Ties-Merging | 0.0328 | 0.8177 | 0.6838 | 0.8284 | 0.7433 | 0.6957 | 0.4765 | 0.6112 |
| TSV-M | 0.0917 | 0.8601 | 0.5882 | 0.8567 | 0.7521 | 0.6464 | 0.5415 | 0.6195 |
| CART | 0.1143 | 0.8624 | 0.5466 | 0.8177 | 0.7010 | 0.7620 | 0.5235 | 0.6182 |
| AdaMerging | 0.0587 | 0.7982 | 0.7083 | 0.8104 | 0.6845 | 0.6758 | 0.4621 | 0.5997 |
| TA+**AdaRank** | 0.0617 | 0.8819 | 0.6275 | 0.7935 | 0.7539 | 0.8131 | 0.4982 | **0.6328** |
| TSV-M+**AdaRank** | 0.1504 | 0.8830 | 0.6985 | 0.8373 | 0.7745 | 0.8168 | 0.5596 | **0.6743** |
| CART+**AdaRank** | 0.1723 | 0.8819 | 0.6544 | 0.7861 | 0.7412 | 0.8437 | 0.5307 | **0.6587** |

