# OpenReview forum: "AdaRank: Adaptive Rank Pruning for Enhanced Model Merging"
_ICLR.cc/2026/Conference — ICLR 2026 Poster_

### Official Review · Reviewer_cS7t · 2025-10-30

**Soundness:** 3
**Presentation:** 3
**Contribution:** 2
**Rating:** 4
**Confidence:** 3

**Summary:**

This paper identifies the suboptimality of direct rank pruning and fixed number of ranks in SVD-based merging. They introduce AdaRank merging technique which dynamically selects top singular components that minimize inter-task interference, and demonstrate its effectiveness through broad settings.

**Strengths:**

1. Overall this paper is highly systematic, written with a good logical flow. The proposed method is tightly motivated by concrete experimental findings on top basis and ranks.
2. Proposed AdaRank is straightforward, lightweight, and compatible with many existing SVD-based methods. Useful and easy for adoption.
3. Main experiment (section 6.2) is broad and covers diverse settings, including different numbers of tasks, datasets, and architectures. Merging methods in comparison are very comprehensive.
4. Solid analysis and supporting materials. Ablation analysis (section 6.3) is well designed and provides a better understanding of the AdaRank method, as well as the behavior of ranks and cross-task singular basis interaction in general and qualitative analysis on the AdaRank’s masks. Further transparency on additional baseline comparison (Appendix D.3) is also appreciated. (though shouldn’t it be included in the main text as well?)

**Weaknesses:**

1. The motivational analysis on top singular components and ranks (section 3) are limited: 1) The experiment is only done on a single ViT model with a specific task set, making the generalizability of this finding toward other models, task types, and training methods unclear. 2) The analysis focuses solely on quantifying the effect of manipulating the top basis and rank, without contextualizing with other relevant information (e.g. how similar are the top singular components across different tasks to begin with, or, how much performance is when preserving only the intrinsic rank). This makes it difficult to make sense of the paper’s analysis in broader perspective. 3) By not always selecting top rank vectors, individual task’s performance would be degraded a lot more than usual, however the paper doesn’t quantify this tradeoff in isolation.
2. Time consumption and computation cost for using AdaRank on modern large-scale models like LLMs is not presented. I suspect that for larger models, the TTA time and memory requirements would be prohibitive.

**Questions:**

1. The comparison with AdaMerging on language models seems unfair.
2. The critical choice of minimizing Shannon entropy is not validated empirically. Its optimality isn’t specifically confirmed for AdaRank, other possible proxies are also not discussed.
3. Table 1 shows that under many settings, merging 20 tasks yields better performance than 14 tasks. Why so?
4. How is it  possible that Oracle achieves performance equal to the Individual model in ablation Figure 4? Also, what’s the models and tasks used?
5. Do the authors plan to discuss or experiment with more recent SVD-based methods (e.g. ISO [1])?
6. AdaRank builds up from the individual SVD paradigm, then how is it related to or differed from joint SVD paradigm (e.g. KnOTS [2], DRM [3])?
7. Does the finding in this paper corroborate with the result from other neighboring adaptive rank-based approaches like AdaLoRA [4]?

[1] https://arxiv.org/pdf/2502.04959v3
[2] https://arxiv.org/pdf/2410.19735
[3] https://arxiv.org/pdf/2505.23117
[4] https://arxiv.org/pdf/2303.10512

---

> ### Author Response · Authors · 2025-11-21
> **Rebuttal for Reviewer cS7t [1/4]**
>
> We thank the reviewer for the insightful questions. We address the raised concerns and questions below.
>
> ---
> ## Weaknesses
>
> > **W1.** Limited Analysis on the Motivation of AdaRank
>
> We appreciate the reviewer’s detailed comments. We updated the **Appendix D.1** of our manuscript by adding (i) loss analyses on diverse models and task sets and (ii) an extended analysis. We provide a summary of the key clarifications.
>
> #### **Additional Analysis across Architectures and Modalities.**
> To address the concern regarding generalizability, we replicated the loss analysis on RoBERTa-base and GPT-2 (Appendix **D.1**, Figure **6**). These models differ fundamentally from ViT: RoBERTa is a bidirectional encoder, while GPT-2 is an autoregressive decoder, and both are trained on text classification data. Despite these distinct pretraining objectives, architectures, and fine-tuning tasks, we observe a consistent trend: the top-$k$ components from the task vector often increase the net multi-task loss when added to a model parameter with other tasks present. This confirms that our finding on the suboptimality of the top-$k$ truncation is a general phenomenon across diverse model merging scenarios, not limited to ViT with image classification tasks.
>
> #### **Why can top singular components be harmful?**
> For further clarification, we add an extended analysis of the nature of the harmful top singular components. First, we clarify that we do not claim that the top singular components always cause inter-task interference. Rather, we argue that rigid top-$k$ selection could be suboptimal because it indiscriminately includes high-magnitude singular components, where any interference results in a substantially larger loss penalty compared to bottom components.
>
> Consider adding an arbitrary singular component from task $i$, denoted as $\mathbf{s}\_{i} = \sigma\_{i} \mathbf{u}\_{i} \mathbf{v}\_{i}^\top$, to the model parameter $\theta$. $L(\theta)=\sum\_{i=1}^TL\_i(\theta)$ is the multi-task loss of $T$ tasks.
> We rewrite the loss change $\Delta L= L(\theta+ \mathbf{s}\_{i})-L(\theta)$ in terms of the direction of the component $\bar{\mathbf{s}}\_{i}=\mathbf{u}\_{i}\mathbf{v}^{\top}\_{i}$ and the singular value $\sigma\_{i}$, using a second-order Taylor expansion:
>
> $\Delta L \approx \sigma\_{i} \nabla L^\top \mathbf{\bar{s}}\_{i} + \frac{1}{2} \sigma\_{i}^2 \mathbf{\bar{s}}\_{i}^\top \mathbf{H} \mathbf{\bar{s}}\_{i},$
>
> where $\mathbf{H}$ denotes the Hessian at $\theta$.
> As seen in the equation, the value of $\Delta L$ is determined by both (i) **directional alignment** between the singular component and the loss curvature, and (ii) the **scale** of the singular value $\sigma\_{i}$. Specifically, the direction decides the sign, and the singular value decides the magnitude of the loss change. We define an "interfering" component as any component where this net change is positive. We showed the value of $\Delta L$ as “Multi-Task Loss Change” plot in Figure **1**, where it implicitly contains both the effects from scale and direction of the singular component.
>
> To decouple these two effects, we compare loss changes when adding (i) the scaled component $\mathbf{s}\_{i}$ (Appendix **D.1**, Figure **7** left) and (ii) component ${\bar{\mathbf{s}}}\_{i}$ without singular values (Figure **7** right). Excluding the effect of the magnitude reveals a uniform distribution of interfering components across the spectral indices, indicating that **the directional misalignment does not differentiate between the interference caused by the top and bottom components**. However, restoring **singular values amplifies the change induced by top components due to their large $\sigma$**: while beneficial ones now help more, misaligned ones cause larger increases in loss.
>
> Overall, these observations explain why fixed top-$k$ selection may include largely interfering singular components, **since any misaligned direction in the range can be amplified by the large singular values**. The existence of these components is likely to make the **top-$k$ subspace suboptimal** under multi-task loss. Pruning only these interfering components in the top-$k$ range (see Table **4**) yields a large performance gain, which empirically demonstrates the suboptimality.

---

> ### Author Response · Authors · 2025-11-21
> **Rebuttal for Reviewer cS7t [2/4]**
>
> #### **Similarity of top singular components.**
> Finally, regarding the similarity of top singular directions: We acknowledge that if tasks are highly similar, their top singular components may align well, making top-$k$ truncation still effective. However, in typical model merging scenarios, our analysis demonstrates that misaligned top components are common. In such cases, a rigid top-$k$ rule is fragile because even few directional misalignments in the top-$k$ range will drive large interference between those components. Importantly, **AdaRank does not exclude the top-$k$ solution; its search space is a superset of it**. If the top-$k$ subspace were indeed optimal, AdaRank could learn to select it. However, as evidenced by our optimization results (Figure 4), AdaRank consistently deviates from the top-$k$ initialization to minimize loss, empirically proving that a superior, non-contiguous subspace exists.
>
> We hope our additional explanation has addressed the reviewer’s concerns regarding our motivational analysis, and we would be happy to further address any remaining ambiguities.
>
> > **W2.** Time / Computational Cost of AdaRank applied on LLMs
>
> We appreciate the reviewer for the helpful comment. We additionally applied AdaRank to merge LLaMa-3.2-3B models (Hidden dim=3072, 32 layers) fine-tuned on three generative tasks: Instruction Following, Code Generation, and Math Solving, following MergeBench protocol [1]. Each task is evaluated with corresponding benchmarks of IFEval, MBPP, and GSM8k. Table A and B summarize the results: AdaRank consistently improves performance across all tasks and outperforms Adamerging. This confirms the applicability of AdaRank to models with large dimensions and parameters, still introducing only a small amount of learnable parameters (**0.015%**). Moreover, as reported in Table B, the SVD cost remains marginal (**~8min**), and the entire TTA procedure finishes within **2 hours** on a single GPU. Given that (i) these costs are incurred only once before deployment, and (ii) they are negligible compared to training or fine-tuning of LLM, we believe **the overhead is reasonable and does not hinder practical application**.
>
> Table **A**. Performance of AdaRank on merging LLaMa3.2-3B models.
> |  |  | Instruction(IFEval) | Math (GSM8k) | Code(MBPP) |
> |---|---|:---:|:---:|:---:|
> | **Individual (Finetuned)** | Instruction | **42.7** | 24.2 | 35.0 |
> |  | Math | 22.2 | **68.8** | 27.4 |
> |  | Code | 18.5 | 24.0 | **39.6** |
> | **Merged** | TA | 37.3 | 51.8 | 40.8 |
> |  | TA+AdaMerging | 39.3 | 53.8 | 41.6 |
> |  | TA+**AdaRank** | **40.2** | **55.7** | **42.2** |
> |  | CART | 40.9 | 57.0 | 40.8 |
> |  | CART+AdaMerging | 40.8 | 58.1 | 39.8 |
> |  | CART+**AdaRank** | **41.7** | **59.7** | **42.0** |
>
> Table **B**. Adaptation-Time Cost of AdaRank on merging LLaMa3.2-3B models.
> |  | Total Parameters| Learnable Parameters | SVD Time | TTA Time |
> |---|:---:|:---:|:---:|:---:|
> | AdaMerging | 9.63B | 765 | - | 98.45min |
> | AdaRank | 9.63B | 1462272 (0.015%) | ~7.6min | 110.2min |
>
> ---
> ## Questions
>
> > **Q1.** The comparison with AdaMerging on language models seems unfair.
>
> Thanks for pointing out. We show the omitted results on RoBERTa and GPT-2 experiments (CART+Adamerging, TSV+Adamerging). We updated the table accordingly. See Table **C** for the results.
>
> **Table C**. Comparison of 7 NLP tasks with merged RoBERTa and GPT-2.
> | Method | RoBERTa | GPT-2 |
> |---|---|---|
> | TA+AdaMerging | 0.6762 | 0.5994 |
> | TA+**AdaRank** | **0.7032** | **0.6328** |
> | TSV-M+AdaMerging | 0.7065 | 0.6219 |
> | TSV-M+**AdaRank** | **0.7309** | **0.6743** |
> | CART+AdaMerging | 0.6954 | 0.6207 |
> | CART+**AdaRank** | **0.7547** | **0.6587** |
>
> > **Q2.** The critical choice of minimizing Shannon entropy is not validated empirically. Its optimality isn’t specifically confirmed for AdaRank, other possible proxies are also not discussed.
>
> We utilized Shannon entropy as our test-time adaptation (TTA) signal primarily because it is the most established and suitable metric for the classification tasks evaluated in our work. Minimizing Shannon entropy is a standard practice in both Test-Time Adaptation literature (e.g., TENT [2], SAR [3]) and adaptive model merging literature (e.g., AdaMerging [4], Twin-Merging [5], WeMOE [6]).
> As the reviewer suggests, different tasks may benefit from other proxies. For example, in an autoregressive language generative model, we could minimize the entropy of the output vocabulary distribution to enhance the certainty of the next-token prediction, and LLM merging results from **W2** are also obtained using this objective. If a pre-merged reference model or a reward model is available, one could minimize the perplexity [7] or use alignment objectives similar to those in RLHF [8], which would be an interesting future work direction.

---

> ### Author Response · Authors · 2025-11-21
> **Rebuttal for Reviewer cS7t [3/4]**
>
> > **Q3.** Table 1 shows that under many settings, merging 20 tasks yields better performance than 14 tasks. Why so?
>
> We apologize for the confusion. We clarify that the average performance does not always decrease monotonically as the number of tasks increases; it depends on which tasks are included in each task configuration. In our setup, the newly added tasks in the 20-task benchmark include several relatively easy tasks with high individual accuracies, which raise the overall average performance. As shown in Table 1, the average individual performance over the 20 tasks is already higher than that of the 14-task set. Therefore, provided that the merged model effectively recovers the pre-merging performance of individual tasks, the average accuracy for 20 tasks is naturally expected to surpass that of the 14 tasks. We will add this clarification to the experimental settings section.
>
> > **Q4.** How is it possible that Oracle achieves performance equal to the Individual model in ablation Figure 4? Also, what’s the models and tasks used?
>
> We apologize for the confusion. The experiment in Figure 4 is conducted on our 8-task vision benchmark using ViT-B/32 as the backbone. We clarify that the **Oracle** in Figure **4** is not intended as a method to be compared in Tables 1 and 2, but as an analysis-only upper bound. Specifically, for the same search space of binary masks over singular components, we construct an Oracle model by optimizing these masks with full access to ground-truth labels on the test set, i.e., by directly minimizing the supervised multi-task loss on the test data. Since it leverage the test labels, the Oracle can match or even slightly exceed the performance of the individually fine-tuned models. Instead, we used Oracle in Figure 4 as a diagnostic tool: First, since the optimization of Oracle is with only respect to the binary masks initialized with top-k selection, consistent reduction in loss indicates that **there exists a strictly better subspace than fixed top-k components**. Second, the figure showed that entropy minimization follows the training curve of the Oracle reasonably well, supporting **entropy as an effective TTA objective for selecting singular components**. We will clarify this in the revised manuscript.
>
> > **Q5.** Do the authors plan to discuss or experiment with more recent SVD-based methods (e.g. ISO)
>
> We appreciate the reviewer for suggesting an additional baseline. We have show the performance of ISO-C and ISO-CTS under our fine-tuned checkpoints, on merging different numbers of tasks. Importantly, ISO-CTS also relies on top-k truncation to build its task-specific subspaces; thus, AdaRank is directly applicable to ISO-CTS. Table **D** summarizes the results: AdaRank consistently improves performance when applied to ISO-CTS across all task configurations. In particular, for the 8-tasks ISO-CTS+AdaRank achieves the best performance among all baselines, exceeding CART+AdaRank (89.2%). This highlights that AdaRank effectively preserves the initial advantages of the base merging methods.
>
> **Table D**. Performance of Isotropic merging on ViT-B-32.
> |ViT-B/32|8 Tasks|14 Tasks|20 Tasks|
> |---|---|---|---|
> |Iso-C|82.8| 78.8 | 73.3 |
> |Iso-CTS | 84.9 | 80.6 | 77.2 |
> |Iso-CTS+**AdaRank**|**89.4 (+4.5)**|**86.8 (+6.2)**|**86.5 (+9.3)**|

---

> ### Author Response · Authors · 2025-11-25
> **Rebuttal for Reviewer cS7t [4/4]**
>
> > **Q6.** AdaRank builds up from the individual SVD paradigm, then how is it related to or differed from joint SVD paradigm (e.g. KnOTS, DRM)
>
> While KnOTS, DRM, and AdaRank all utilize SVD; their objectives and formulations differ significantly. KnOTS and DRM perform a joint SVD over concatenated task vectors to construct a shared basis across tasks. Their goal is to align the output space or representations of different tasks before merging by explicitly enforcing structure within this shared subspace. In contrast, AdaRank operates on individual SVDs per task vector and does not attempt to build a shared basis. Instead, our focus is on selecting a subset of singular components for each task that minimizes inter-task interference in the merged model. These two approaches are complementary directions, and could potentially be combined for further improvement.
>
> > **Q7.** Does the finding in this paper corroborate with the result from other neighboring adaptive rank-based approaches like AdaLoRA?
>
> AdaLoRA shares our motivation of adaptively finding beneficial low-rank subspaces, and we corroborate its finding that dynamic rank distribution across layers is essential. However, the problem setting and implications differ. AdaLoRA targets single-task fine-tuning where inter-task interference is absent; thus, selecting top components remains optimal for reconstruction. In contrast, model merging must address inter-task interference driven by those top components. Consequently, unlike AdaLoRA, AdaRank does not strictly retain top-k components but identifies a **mixture** that prunes certain interfering top components while retaining beneficial ones from the **lower spectrum**. (See Figure **9** in the Appendix).
>
>
> ---
> [1] He et al. “MergeBench: A Benchmark for Merging Domain-Specialized LLMs”, NeurIPS 2025.
>
> [2] Wang et al. “Tent: Fully Test-Time Adaptation by Entropy Minimization.” ICLR 2021.
>
> [3] Niue et al. “Towards Stable Test-Time Adaptation in Dynamic Wild World”, ICLR 2023.
>
> [4] Yang et al. "Adamerging: Adaptive model merging for multi-task learning." ICLR 2023.
>
> [5] Lu et al. “Twin-merging: Dynamic integration of modular expertise in model merging.” NeurIPS 2024.
>
> [6] Tang et al. “Merging multi-task models via weight-ensembling mixture of experts.” ICML 2024
>
> [7] Hu, Jinwu, et al. "Test-Time Learning for Large Language Models." ICML 2025
>
> [8] Ouyang, Long, et al. "Training language models to follow instructions with human feedback." NeurIPS 2022

---

### Official Review · Reviewer_o9CP · 2025-11-02

**Soundness:** 2
**Presentation:** 3
**Contribution:** 2
**Rating:** 4
**Confidence:** 4

**Summary:**

The paper addresses the problem of test-time adaptation for model merging, which involves learning a set of parameters on an unsupervised set to perform the merging of models. Since the task matrices exhibit varying effective ranks across tasks and network layers, using a fixed rank in SVD-based merging techniques—by selecting a constant number of top-$k$ components—can be suboptimal. To overcome this limitation, the authors propose an adaptive mechanism based on binary masks, which dynamically selects the singular components for each task matrix.

Specifically, the proposed algorithm, AdaRank, builds upon a generic data-free merging method operating on task matrices (such as Task Arithmetic, CART, or TSV). It first computes the task matrix as specified by the chosen method, then performs an SVD decomposition of the resulting task matrices for each task and layer. These binary masks are applied to the singular values and optimized during training via the Straight-Through Estimator (STE),  together with learnable task coefficients, with the objective of minimizing the entropy on the unsupervised set. The approach is evaluated on both vision and language tasks.

**Strengths:**

- The empirical findings showing that selecting only the top singular components of task vectors introduces interference is clearly presented. I particularly appreciated the experiments in Section 3, which illustrate how the loss evolves as singular components are progressively added to the merged task matrix.
- The proposed approach significantly improves performance over AdaMerging, the previous method that learns task coefficients for model merging on the unsupervised set.
- Extensive experiments are conducted to validate the effectiveness of the method.

**Weaknesses:**

-  From a methodological perspective, the novelty of the proposed approach is limited. The unsupervised loss function is identical to that used in AdaMerging. The authors introduce binary masks, which were already employed in TALL [1], although in that work they were not learnable and were applied to task matrices. The use of the Straight-Through Estimator (STE), as acknowledged by the authors themselves, is also a well-established technique. Overall, the method appears to combine existing components in a reasonable way, but without introducing a significant methodological contribution.

- While the empirical findings showing that adding top components introduces interference are interesting, the paper does not provide a clear explanation or theoretical insight into why this phenomenon occurs (see the Question section).

- As the number of tasks, network layers, and the dimensionality of the feature space increase, the computational complexity of the approach also grows. It is unclear whether the method remains effective under these conditions (see the Question section).

[1] Wang, K., Dimitriadis, N., Ortiz-Jiménez, G., Fleuret, F., & Frossard, P. (2024). Localizing task information for improved model merging and compression. In Proceedings of the International Conference on Machine Learning (ICML).

**Questions:**

Q1) Why does adding top principal directions introduce interference? Is this effect driven by the scale of the singular values or by the specific directions of the singular vectors? Moreover, since task singular vectors interact with one another (while the experiments only show what happens when a single task singular vector is added at a time), it would be interesting to analyze how interference emerges from the joint combination of singular vectors. Finally, the task singular vectors also interact with those of the pre-trained backbone. It remains unclear how the top singular vectors of each task matrix align or interfere with the singular directions of the pre-trained network, and whether the proposed method mitigates any of these effects.

Q2) I have seen the computational analysis performed in the Appendix on ViT-B/32 and ViT-L/14. However, for recent large language models, it may not be true that the additional overhead introduced by the method remains small compared to AdaMerging. These models have many more layers, and, more importantly, their feature space dimensionality is significantly larger than that of ViTs, leading to very large binary masks. Have the authors tested how the method scales when applied to some LLM with higher feature dimensionality? Is binary masking still effective in such settings? And does adding the top singular vectors remain detrimental as shown in Figure 1?

Overall, as mentioned in the weaknesses, the proposed method mainly builds upon existing strategies (the entropy minimization, the STE estimator and the binary masks). However, since using learnable binary masks on the singular values of the network achieves good empirical results, the paper would benefit from providing stronger theoretical or empirical justification—particularly to explain the interference caused by the top singular vectors (as raised in Q1), which I consider more important than the computational and performance scalability issues discussed in Q2.

---

> ### Author Response · Authors · 2025-11-21
> **Rebuttal for Reviewer o9CP [1/3]**
>
> We thank the reviewer for the detailed methodological analysis and insightful questions. We address the raised concerns and questions below.
>
> ---
> ## Weaknesses
>
> > **W1.** Lack of Methodological Novelty
>
> We appreciate the reviewer’s careful analysis and agree that our method builds upon components such as entropy-based TTA, binary masking, and the STE estimator.
> However, the novelty of AdaRank does not lie in introducing these components individually, but in resolving a core methodological limitation shared across all prior SVD-based merging methods.
>
>
> Prior SVD-based approaches [1,2,3,4,5] implicitly assume that the top-k singular components form the optimal low-rank subspace for model merging. This assumption has never been questioned or relaxed in the literature. Our analyses, to the best of our knowledge, provide the **first systematic evidence** that this assumption is not valid in multi-task settings where top singular components often introduce large inter-task interference.
>
>
> To address this, AdaRank introduces per-component binary masks and apply test-time adaptation (TTA) to discover the combination of singular components that minimize the inter-task interference. Despite the simplicity, we believe that this is the most direct and effective solution to the problem we discovered in the paper. We found that such simplicity is advantageous in various aspects, since it allows only marginal TTA cost (**~0.03%** of total parameters), data efficiency (e.g., significant improvement by using only **1%** of test data; see our response to **W1** of reviewer `mGGd`), and scalability to large models such as LLMs (see our response to **W3**).
>
>
> Overall, we believe AdaRank as a concise but conceptually important approach that revisits a widely adopted top-$k$ assumption in SVD-based model merging and provides a simple mechanism that consistently improves diverse existing methods. We believe these findings provide meaningful insights and contributions to the model merging community and hope the reviewer considers these contributions.
>
> ---
>
> > **W2 / Q1.** Lack of explanations on Top Component Interference
>
> We thank the reviewer for raising this question and agree that a clearer explanation is needed. We updated our Appendix (see Section **D.1**) with extended analysis and corresponding experiments. Below, we summarize the key findings.
>
>
> #### **Condition for interference.**
> We clarify that we do not claim that the top singular components always cause inter-task interference. Rather, we argue that rigid top-$k$ selection could be suboptimal because it indiscriminately includes high-magnitude singular components, where any interference results in a substantially larger loss penalty compared to bottom components.
>
> Consider adding an arbitrary singular component from task $i$, denoted as $\mathbf{s}\_{i} = \sigma\_{i} \mathbf{u}\_{i} \mathbf{v}\_{i}^\top$, to the model parameter $\theta$. $L(\theta)=\sum\_{i=1}^TL\_i(\theta)$ is the multi-task loss of $T$ tasks.
> We rewrite the loss change $\Delta L= L(\theta+ \mathbf{s}\_{i})-L(\theta)$ in terms of the direction of the component $\bar{\mathbf{s}}\_{i}=\mathbf{u}\_{i}\mathbf{v}^{\top}\_{i}$ and the singular value $\sigma\_{i}$, using a second-order Taylor expansion:
>
> $\Delta L \approx \sigma\_{i} \nabla L^\top \mathbf{\bar{s}}\_{i} + \frac{1}{2} \sigma\_{i}^2 \mathbf{\bar{s}}\_{i}^\top \mathbf{H} \mathbf{\bar{s}}\_{i},$
>
> where $\mathbf{H}$ denotes the Hessian at $\theta$.
> As seen in the equation, the value of $\Delta L$ is determined by both (i) **directional alignment** between the singular component and the loss curvature, and (ii) the **scale** of the singular value $\sigma\_{i}$. Specifically, the direction decides the sign, and the singular value decides the magnitude of the loss change. We define an "interfering" component as any component where this net change is positive. We showed the value of $\Delta L$ as “Multi-Task Loss Change” plot in Figure **1**, where it implicitly contains both the effects from scale and direction of the singular component.
>
> #### **Singular Value Magnitude vs. Directional Alignment.**
>
> As asked by the reviewer, we now investigate the primary cause driving the interference in the top components. To decouple these two effects, we compare loss changes when adding (i) the scaled component $\mathbf{s}\_{i}$ (Appendix **D.1**, Figure **7** left) and (ii) component ${\bar{\mathbf{s}}}\_{i}$ without singular values (Figure **7** right). Excluding the effect of the magnitude reveals a uniform distribution of interfering components across the spectral indices, indicating that **the directional misalignment does not differentiate between the interference caused by the top and bottom components**. However, restoring **singular values amplifies the change induced by top components due to their large $\sigma$**: while beneficial ones now help more, misaligned ones cause larger increases in loss.

---

> ### Author Response · Authors · 2025-11-21
> **Rebuttal for Reviewer o9CP [2/3]**
>
> #### **Joint effect.**
> As the reviewer mentioned, the joint interaction between different singular components may induce additional effects on the loss. Consider simultaneously adding two singular components $\mathbf{s}\_i$ and $\mathbf{s}\_j$, each from task $i$ and $j$, to $\theta$. Using a second-order Taylor expansion, the multi-task loss change $\Delta L = L(\theta + \mathbf{s}\_i + \mathbf{s}\_j) - L(\theta)$ can be expressed as:
>
> $\Delta L \approx \sum\_{p \in \{ i,j \} } \left( \sigma\_p \nabla L^\top \mathbf{\bar{s}}\_p + \frac{1}{2} \sigma\_p^2 \mathbf{\bar{s}}\_p^\top \mathbf{H} \mathbf{\bar{s}}\_p \right) + \sigma\_i \sigma\_j (\mathbf{\bar{s}}\_i^\top \mathbf{H} \mathbf{\bar{s}}\_j),$
>
> where $\bar{\mathbf{s}}\_p=\mathbf{u}\_p\mathbf{v}\_p^{\top}$. The terms inside the summation correspond to the loss changes from adding each component individually and are independent of the other component. The joint interaction arises from the last term, which is a Hessian inner product between two directions. As in the single-component case, this interaction is governed by (i) the directional alignment of $\bar{\mathbf{s}}\_i$ and $\bar{\mathbf{s}}\_j$ with respect to the loss curvature, and (ii) the magnitudes of the singular values $\sigma\_i$ and $\sigma\_j$.
> Empirically, we quantify this joint interaction by measuring:
>
> $\Delta L_{\text{joint}} = L(\theta + \mathbf{s}\_i + \mathbf{s}\_j) - (L(\theta + \mathbf{s}\_i) + L(\theta + \mathbf{s}\_j) - L(\theta)),$
>
> which isolates the additional loss change that occurs only when $\mathbf{s}\_i$ and $\mathbf{s}\_j$ are added together.
>
> To see how top components interact with each other, we fix $\mathbf{s}\_i$ as the top-1 component from task $i$, and sweep over all singular components of task $j$ to measure this joint interaction. Figure **8** summarizes the results: we observe a similar trend as in the single-component analysis; the directional interaction does not show a strong dependence on the index of $\mathbf{s}\_j$, but **interactions involving top components become much larger** after accounting for their singular values. The joint effect is most amplified when both $\mathbf{s}\_i$ and $\mathbf{s}\_j$ originate from the top range, where their large singular values quadratically scale the interaction term.
> Therefore, joint interaction reinforces the same conclusion as the single-component addition: **the risk of interference arises from misaligned directions being amplified by large singular values, especially for those prominent in the top range**.
>
> Overall, these observations explain why fixed top-$k$ selection may include largely interfering singular components, **since any misaligned direction in the range can be amplified by the large singular values**. The existence of these components is likely to make the **top-$k$ subspace suboptimal** under multi-task loss. Pruning only these interfering components in the top-$k$ range (see Table **4**) yields a large performance gain, which empirically demonstrates the suboptimality.
>
> #### **Interaction with Pre-Trained Backbone.**
> We thank the reviewer for pointing out the potential interaction between task singular directions and those of the pre-trained backbone. However, we clarify that in this paper, we define **“interference”** purely as the multi-task loss change resulting from adding task vector’s singular components. In this sense, interference is a relation between task vectors, measuring how a singular component affects the losses of other tasks. Since the pre-trained backbone serves as a fixed origin, there is no well-defined notion of “interference between a task and the backbone” under our definition. Therefore, AdaRank and our analysis focus on how singular components interfere with each other given a fixed backbone, rather than on how they align with the backbone’s own singular directions. If we have misinterpreted the reviewer’s intent, however, we would be happy to engage in further discussion to clarify this point.

---

> ### Author Response · Authors · 2025-11-21
> **Rebuttal for Reviewer o9CP [3/3]**
>
> > **W3 / Q2.** Scalability to Large Language Models (LLMs)
>
> To further demonstrate scalability, we additionally applied AdaRank to merge LLaMa-3.2-3B models fine-tuned on three domains of generative tasks: instruction-following, code generation, and math solving, following MergeBench [6]. The model has hidden dim=3072 with 32 attention blocks, which are **3x** and **1.5x** bigger than our largest experiment (ViT-L/14). Each task is evaluated with corresponding benchmarks of IFEval, MBPP, and GSM8k.
> Tables **A** and **B** summarize the results: AdaRank consistently improves performance across all tasks and outperforms Adamerging. This confirms the applicability of AdaRank to models with large dimensions and parameters,, and in particular, improvement over CART (top-16% intialized) shows that the suboptimality of top-k truncation still exists in task vectors with large-dimensional matrices.
> Moreover, as reported in Table **B**, AdaRank still introduces only a small portion of learnable parameters compared to the full set (**0.015%**). In terms of time cost, the SVD cost remains marginal (**~8min**), and the entire TTA procedure finishes within **2 hours** on a single GPU, making the total time cost comparable to AdaMerging. Given that (i) these costs are incurred once before deployment, and (ii) they are negligible compared to training or full fine-tuning LLM, we believe **the overhead is reasonable and does not hinder practical application**.
>
> **Table A**. Performance of AdaRank on merging LLaMa3.2-3B models.
> |  |  | Instruction(IFEval) | Math (GSM8k) | Code(MBPP) |
> |---|---|:---:|:---:|:---:|
> | **Individual (Finetuned)** | Instruction | **42.7** | 24.2 | 35.0 |
> |  | Math | 22.2 | **68.8** | 27.4 |
> |  | Code | 18.5 | 24.0 | **39.6** |
> | **Merged** | TA | 37.3 | 51.8 | 40.8 |
> |  | TA+AdaMerging | 39.3 | 53.8 | 41.6 |
> |  | TA+**AdaRank** | **40.2** | **55.7** | **42.2** |
> |  | CART | 40.9 | 57.0 | 40.8 |
> |  | CART+AdaMerging | 40.8 | 58.1 | 39.8 |
> |  | CART+**AdaRank** | **41.7** | **59.7** | **42.0** |
>
> **Table B**. Adaptation-Time Cost of AdaRank on merging LLaMa3.2-3B models.
> |  | Total Parameters| Learnable Parameters | SVD Time | TTA Time |
> |---|:---:|:---:|:---:|:---:|
> | AdaMerging | 9.63B | 765 | - | 98.45min |
> | AdaRank | 9.63B | 1462272 (0.015%) | ~7.6min | 110.2min |
>
> ---
> [1] Choi et al. “Revisiting Weight Averaging for Model Merging.” arXiv 2024.
>
> [2] Gargiulo et al. “Task singular vectors: Reducing task interference in model merging.” CVPR 2025.
>
> [3] Marczak et al. “No Task Left Behind: Isotropic Model Merging with Common and Task-Specific Subspaces.” ICML 2025.
>
> [4] Lee et al. “STAR: Spectral Truncation and Rescale for Model Merging.” NAACL 2025
>
> [5] Lu et al. “Twin-merging: Dynamic integration of modular expertise in model merging.” NeurIPS 2024.
>
> [6] He et al. “MergeBench: A Benchmark for Merging Domain-Specialized LLMs”, NeurIPS 2025.

---

> ### Comment · Reviewer_o9CP · 2025-11-26
>
> Thanks to the authors for the detailed answer to my questions.
>
> First, regarding the interaction with pre-trained backbones, let me clarify my point. A pre-trained backbone can also be decomposed via SVD, $\theta = U \Lambda V$, where $U$ and $V$ span the output and input spaces of the layer’s weight. I wondered wheter the reported multi-task loss change arise from interactions between the principal directions of task matrices and those of the backbone, specifically the activation-related directions captured by $U$.  This perspective does not seem far from what the authors discuss in the rebuttal regarding the alignment of singular directions with the Hessian provided in this rebuttal (which depends on the backbone weights). Given this, I accept the definition of interference provided by the authors.
>
> I kindly ask the authors to provide some additional clarifications about the other points in the rebuttal:
>
> - *Appendix D experiment*. I feel that the analyses on singular value magnitude vs directional alignment are on the right track, but the conclusion that "directional effect does not distinguish the interference cause by top and bottom components" is not fully supported. In Figure 7 right plot, there is a still clear bump in the loss for the top components, similar to the left plot. While I agree that the middle and bottom components show more uniform noise, the top directions still appear to introduce interference regardless of scale.
>
> I also note that the right plot uses a different vertical scale from the left. A key issue with the experiment is that setting all singular values to 1 artificially change the normal behaviour of the network, amplifying small noisy components. A more appropriate way to separate directional alignment from spectral scale would be to define:
>
> $$ \overline{s}_{ir} = \overline{\sigma} u_{ir} {v_{ir}}^\top $$ (Figure 7, right)
>
> where $\overline{\sigma}$ is estimated from the real spectrum, such as an average excluding the bottom noisy singular values. In this way, the authors can clearly distinguish between directional alignment and spectral scale. The same observation applies to Figure 8.
>
> - On the hessian. Since the authors introduced in this rebuttal the Hessian, I would appreciate clarification on whether interference between singular directions and the Hessian can be analyzed. Beyond the mathematical elegance of using the Hessian to describe multi-task loss changes, studying how singular directions interact with the Hessian’s principal directions could shed further light on why top-$k$ components interfere more. I understand that computing full hessian is infeasible for large model, but approximations,e.g. layerwise or last layer Hessian, might make such analysis possibile.
>
> As mentioned in my previous revision, I believe that these  analysis could explain why AdaRank selects directions beyond the top-$k$ (figure 5 (a)). Is task arithmetic the merging strategy used for this plot? I also noticed looking the other rebuttal questions that Reviewer BUd9 asked to evaluate Iso-C, which flattens the singular values. How does this plot look when using Iso-C method ?

---

> > ### Author Response · Authors · 2025-11-29
> > **Response for Reviewer o9CP**
> >
> > We thank the reviewer for the helpful feedback on our analysis.
> >
> > > Correct scale of singular value for the direction-only experiment.
> >
> > We agree that setting values to 1 may artificially upweight small directions. As suggested, we reran the experiment by rescaling all singular components using the mean singular value ($\bar{\sigma}$) of the top 80\% of the spectrum (see Figures **9(a)** and **9(b)**, Appendix D.1). Even with this revised scale, the qualitative pattern remains: directions with $\Delta L > 0$ are not only prominent in the top spectral range.
> >
> > > Alignment between singular directions and the Hessian
> >
> > We examined the interaction between singular directions and the Hessian. However, computing the full Hessian and its principal components is computationally prohibitive under our resource constraint even for a single layer. Therefore, we instead calculated the quadratic term $s_i^\top H s_i$ using Hessian-vector products (HVP). Although this does not give us the explicit principal components of $H$, the value sufficiently captures the aggregate alignment of singular directions with high-curvature regions.
> >
> >
> >
> > We added Figure **9(d)**, which plots $s_i^\top H s_i$ using the same mean-rescaled $\bar{\sigma}$ as above. Interestingly, we observe a distinct pattern compared to the total loss: top singular components tend to exhibit positive curvature ($s_i^\top H s_i > 0$), while many bottom components show negative values. We thus relax our earlier claim that “the directional effect does not distinguish the interference caused by top and bottom components”, since the alignment with the Hessian’s principal directions can differ across the spectrum.
> >
> > At the same time, we argue that this curvature effect is not the primary driver of interference in the top range. The actual loss change in Figure **9(a)**, combining both the first- and second-order contributions under fixed $\bar{\sigma}$, shows no prominent concentration of interfering components in the top range. This observation is further supported by Figure **9(c)**, where we plot the first-order term $\nabla L^\top s_i$ under the same fixed $\bar{\sigma}$. The first-order term exhibits behavior consistent with the actual loss change in Figure 9(a): interfering directions ($\Delta L > 0$) appear across the spectrum rather than being concentrated at the top indices. Taken together, these results imply that alignment with the Hessian does not, by itself, create a strong separation between top and bottom components in the actual loss change (Figure **9(a)**) since the first-order term effects dominates over it, while the mild “bump’’ at the top range noted by the reviewer can be partly attributed to alignment with the upward curvature.
> >
> > To conclude our analysis on top-$k$ interference: we decomposed the loss change into first-order ($\nabla L^\top s_i$), quadratic ($s_i^\top H s_i$), and joint terms, and under fixed $\bar{\sigma}$, we found that only the quadratic term shows more interfering terms in top range, while the first-order and joint terms produce interfering directions across the spectrum. Overall, these results suggest that the Hessian does introduce a directional
> > effect—top components are more aligned with upward curvature than bottom components—but that this effect is not strong enough to substantially alter the index-wise distribution of interfering components. In this sense, when comparing spectral scale and directional effects, we still view the size of the singular values as the more dominant factor in shaping the top-range interference pattern.
> > .
> > > Figure 5(a) and Iso-C
> >
> > Figure **5 (a)** was generated using TA+AdaRank. For Iso-C, we initially interpreted the question as applying AdaRank on top of Iso-C and see the mask distribution of the results. However, Iso-C is not directly compatible with our AdaRank formulation: Iso-C operates by flattening the spectrum of the **summed** task vector $\sum_i \tau_i$ and does not produce per-task singular components that Figure **5(a)** uses. While Iso-C flattens the singular values of the summed task vector, this does not imply that the singular values of the individual task vectors are also flattened; this differs from the per-task, direction-only analysis we consider in this rebuttal.
> >
> > > On AdaRank’s behavior selecting directions beyond top-k
> >
> > We agree that our curvature analysis partially explains why AdaRank selects directions beyond top-$k$ (Figure **5(a)**). Since bottom components often exhibit lower or negative curvature, they can offer useful updates with reduced interference, thus can be selected by AdaRank. However, as mentioned above, this effect is secondary to the dominant impact of singular-value scaling; where it suppresses beneficial bottom components. Consistent with this, Table 4 confirms that AdaRank’s primary gains come from pruning misaligned top components, while selecting non-top components yields smaller additional improvements.

---

### Official Review · Reviewer_BUd9 · 2025-11-03

**Soundness:** 3
**Presentation:** 3
**Contribution:** 2
**Rating:** 6
**Confidence:** 3

**Summary:**

This work present a novel method for model merging - Adaptive Rank Prunning (AdaRank). AdaRank learns binary mask for merging most useful components from SVD decomposition based on the test data. During the optimization, as a proxy objective the entropy minimization known from test-time adaptation methods is used. Comparing to the fixed SVD rank truncation for visual and language models, AdaRank presents significant improvement.

**Strengths:**

1. Clear motivation with a good analysis for backing the idea that (1) top singular components may benefit one task but harm others and (2) fixed rank truncation fails to account for task- and layer-specific complexity differences.

1. Significant improvements over AdaMerge.

**Weaknesses:**

1. I do not see the point of comparing AdaRank in Fig. 3 and putting it to the main paper. What we what to present show exactly - as the number of parameters with router-base methods are expected to be increasing with the number of task right? Maybe reconsider putting the number of the parameters and optimization of AdaRank in comparison to the AdaMerge for adaptation-time cost section. I found it more useful for presenting AdaRank method.

1. Only single dynamic merging method is compared, and the work do not include more recent strong method, e.g. [1] that is static and present better results for some settings.

1. AdaRank assumes access to the test data without labels, however, here the authors do not present any analysis about how the method performing with proposed entropy minimization with respect to the provided data (number of samples, covariance shift, etc.).

[1] Marczak, D., Magistri, S., Cygert, S., Twardowski, B., Bagdanov, A. D., & van de Weijer, J. (2025). No task left behind: Isotropic model merging with common and task-specific subspaces, ICML 2025

**Questions:**

1. What is the performance of the Oracle model presented in Figure 4 in the experiments presented in Tab 1 and Tab 2?

1. Why in Table 2 we see different composition of the method {TA,TSV-M, CART} x {AdaMerging, AdaRank}?

---

> ### Author Response · Authors · 2025-11-21
> **Rebuttal for Reviewer BUd9 [1/2]**
>
> We appreciate the reviewer's constructive feedback and suggestions for improving the manuscript. Here we address the raised concerns and qustions below.
>
> ## Weaknesses
>
> > **W1.**  Moving the adaptation-time cost section to the main paper
>
> We appreciate the feedback. We updated the manuscript as recommended, by moving the adaptation-time cost section to the main paper.
>
> > **W2.** Only single dynamic merging method is compared, and the work do not include more recent strong method (e.g. ISO) that is static and present better results for some settings.
>
> We appreciate recommending an additional baseline. To clarify, we kindly remind the reviewer that we already compared other dynamic merging methods such as Representation Surgery (post-merging) or Router-based approaches; though they operate under different constraints apart from access to test data, we discussed the settings and compared the performance separately in **Table 4** and **Appendix D.3**.
>
> Regarding Isotropic Merging [1], we clarify that their reported results relied on stronger fine-tuned checkpoints. When reproduced under identical conditions, AdaRank outperforms Iso-CTS. Nonetheless, Iso-CTS is still a static merging method and is compatible with AdaRank since Iso-CTS also relies on top-k truncation to build its task-specific subspaces. Combined with Iso-CTS, we observe a further performance gain (see **Table A**). In particular, for the 8-task benchmark Iso-CTS+AdaRank achieves the best performance among all baselines, exceeding CART+AdaRank (89.2%). This highlights that AdaRank effectively preserves the initial advantages of the base merging methods.
>
> **Table A**. Performance of Iso-C, Iso-CTS, Iso-CTS+AdaRank on ViT-B-32.
> | ViT-B/32 | 8 Tasks | 14 Tasks | 20 Tasks |
> |---|---|---|---|
> | Iso-C | 82.8 | 78.8 | 73.3 |
> | Iso-CTS | 84.9 | 80.6 | 77.2 |
> | Iso-CTS+**AdaRank** | **89.4 (+4.5)** | **86.8 (+6.2)** | **86.5 (+9.3)** |
>
> > **W3**. Robustness of AdaRank’s performance on the test data
>
> We appreciate the reviewer’s comment. Following the suggestion, we conducted an additional experiment varying the available amount of test data. The results are summarized in Tables **B** and **C**. With access to only **1%** of the test set, AdaRank already outperforms AdaMerging, which uses the **full** test dataset. Furthermore, AdaRank with 1% test data still offers significant gains over the baseline models, improving Task Arithmetic by **10.8%** and CART by **3.5%**.
>
> This effectiveness arises because AdaRank does not learn new representations but only decides which existing singular directions should be activated or suppressed. Since each singular component already isolates a task-specific update, the entropy objective only needs to identify a small, structured subset of beneficial directions, avoiding overfitting and allowing the optimization to converge reliably with very few samples.
>
> We appreciate the comment and will include the results in the main paper.
>
> **Table B**. Performance of AdaRank on different amounts of test data, on ViT.
> |ViT-B-32 | Method | 0% (Static) | 1% | 5% | 10% | 50% | 100% | AdaMerging(100%) |
> |:---:|:---:|:---:|:---:|:---:|:---:|:---:|:---:|:---:|
> | 8 Tasks | TA+AdaRank | 69.2 | 81.2 | 84.2 | 85.2 | 87.2 | 87.9 | 80.1 |
> |  | CART+AdaRank | 65.4 | 76.8 | 78.2 | 79.0 | 81.3 | 82.1 | 76.7 |
> | 14 Tasks | TA+AdaRank | 61.0 | 70.0 | 71.3 | 73.0 | 76.7 | 81.4 | 69.2 |
> |  | CART+AdaRank | 84.7 | 86.1 | 87.1 | 87.7 | 88.6 | 89.2 | 85.9 |
> | 20 Tasks | TA+AdaRank | 79.5 | 82.8 | 84.4 | 85.3 | 85.8 | 86.2 | 82.3 |
> |  | CART+AdaRank | 76.8 | 82.7 | 84.3 | 85.2 | 85.8 | 86.4 | 82.7 |
>
> **Table C**. Performance of AdaRank on different amounts of test data, on RoBERTa / GPT-2.
>
> |NLP| Method | 0% (Static) | 1% | 5% | 10% | 50% | 100% | AdaMerging(100%) |
> |:---:|:---:|:---:|:---:|:---:|:---:|:---:|:---:|:---:|
> | RoBERTa | TA+AdaRank | 0.6718 | 0.6774 | 0.6909 | 0.6937 | 0.6947 | 0.7032 | 0.6762 |
> |  | CART+AdaRank | 0.6997 | 0.7197 | 0.7087 | 0.7497 | 0.7453 | 0.7547 | 0.6954 |
> | GPT-2 | TA+AdaRank | 0.5910 | 0.6330 | 0.6270 | 0.6304 | 0.6337 | 0.6328 | 0.5994 |
> |  | CART+AdaRank | 0.6185 | 0.6524 | 0.6484 | 0.6528 | 0.6511 | 0.6587 | 0.6207 |

---

> ### Author Response · Authors · 2025-11-21
> **Rebuttal for Reviewer BUd9 [2/2]**
>
> ## Questions
>
> > **Q1.** What is the performance of the Oracle model presented in Figure 4 in the experiments presented in Tab 1 and Tab 2?
>
> We apologize for the confusion. We clarify that the **“Oracle”** in Figure **4** is not intended as a method to be compared in Tables **1** and **2**, but as an analysis-only upper bound. Specifically, for the same search space of binary masks over singular components, we construct an Oracle model by optimizing these masks with full access to ground-truth labels on the test set, i.e., by directly minimizing the supervised multi-task loss on the test data. Since this leads to memorization of test data, reporting its performance would not be informative. Instead, we used Oracle in Figure **4** as a diagnostic tool: First, since the optimization of Oracle is with only respect to the binary masks initialized with top-k selection, consistent reduction in loss indicates that **there exists a strictly better subspace than fixed top-k components**. Second, the figure showed that entropy minimization follows the training curve of the Oracle reasonably well, supporting **entropy as an effective TTA objective for selecting singular components**. We will clarify this in the revised manuscript.
>
> > **Q2.** Why in Table 2 we see different composition of the method {TA,TSV-M, CART} x {AdaMerging, AdaRank}?
>
> Thanks for pointing out. We show the omitted results on RoBERTa and GPT-2 experiments (CART+AdaMerging, TSV+AdaMerging). We updated Table **2** accordingly. See Table **D** for results.
>
> Table **D**. Comparison of 7 NLP tasks with merged RoBERTa and GPT-2.
> | Method | RoBERTa | GPT-2 |
> |---|---|---|
> | TA+AdaMerging | 0.6762 | 0.5994 |
> | TA+**AdaRank** | **0.7032** | **0.6328** |
> | TSV-M+AdaMerging | 0.7065 | 0.6219 |
> | TSV-M+**AdaRank** | **0.7309** | **0.6743** |
> | CART+AdaMerging | 0.6954 | 0.6207 |
> | CART+**AdaRank** | **0.7547** | **0.6587** |
>
> ---
> [1] Marczak et al. “No Task Left Behind: Isotropic Model Merging with Common and Task-Specific Subspaces.” ICML 2025.

---

### Official Review · Reviewer_mGGd · 2025-11-11

**Soundness:** 3
**Presentation:** 3
**Contribution:** 3
**Rating:** 4
**Confidence:** 5

**Summary:**

This paper introduces a novel model merging framework named **AdaRank (Adaptive Rank Pruning)**. It is designed to overcome two
core limitations associated with heuristic rank selection in existing Singular Value Decomposition (SVD)-based model merging methods:
**inter-task interference** and **suboptimal fixed rank allocation**. The core contribution of AdaRank lies in its departure from the traditional strategy of retaining only the top $k$ largest singular components. Instead, it introduces a **learnable binary mask** for every singular component. To optimize this mask—thereby enabling the automatic selection of beneficial components and pruning of detrimental ones—the authors utilize unlabeled test data and formulate an unsupervised surrogate objective based on entropy minimization. Through the adaptive optimization of these masks, AdaRank effectively assigns an adaptive effective rank to different tasks and model layers, leading to a significant improvement in the performance of the multi-task merged model.

**Strengths:**

- **Fundamental Improvement on SVD-based Merging.** AdaRank is built upon established SVD low-rank approximation theory, but critically, it clearly articulates the limitations of traditional  Top-k heuristic truncation, namely, leading to inter-task interference and
suboptimal fixed rank allocation. The core innovation is replacing this rigid truncation with a learnable, per-component binary mask, enabling a dynamic, adaptive selection of each singular component's retention level. This change is backed by solid theoretical motivation and provides significant flexibility.
- **Practical Solution for Data Scarcity.** Addressing the common challenge of **unavailable training data** in model merging scenarios, the paper ingeniously avoids dependence on the original training set. It achieves this by employing Test-Time Adaptation (TTA) and minimizing the prediction entropy on unlabeled test data as a surrogate optimization objective. This successfully ensures zero reliance on training data while simultaneously bolstering the merged model's prediction confidence and performance.

**Weaknesses:**

- **Introduction of additional computational complexity and time overhead.** Compared to traditional algebraic operation-based model merging methods (e.g., Task Arithmetic, TSV-M), AdaRank introduces an additional test-time adaptation (TTA) optimization phase to learn binary masks. This transforms the merging process from an instantaneous pure algebraic operation into an iterative gradient descent procedure, resulting in significant training resource and time overhead. Furthermore, the method retains the preprocessing step of performing SVD decomposition on task vectors, which itself is a time- and memory-intensive operation when dealing with massive
weight matrices in large-scale models (e.g., LLMs), further limiting its applicability in scenarios that demand extreme efficiency.

- **Gradient approximation issues due to the discrete nature of binary masks.** Since the binary mask B is discrete, the Straight-Through Estimator (STE) strategy is employed for gradient approximation during optimization. This approach may lead to inaccurate gradient approximation and convergence to local minima.

- **Dependence on unlabeled test data limits generalizability.** Although AdaRank avoids dependence on original training data, it still
requires access to unlabeled test datasets. In certain extreme or confidential scenarios where test data is also scarce or completely
unavailable, the entropy minimization optimization step of AdaRank cannot be executed, rendering the method inapplicable. Moreover, if the test sample size is too small, it may not adequately represent the true distribution of the task, leading to suboptimal and less robust mask
configurations.

**Questions:**

- Does the AdaRank method maintain robust performance in extremely low-resource scenarios (where test data is very scarce)? Could the authors provide an ablation study analyzing the relationship between performance and the number of test samples used for entropy minimization? Specifically, determine the minimum sample size required for AdaRank to achieve performance comparable to or better than baseline methods under different task complexity levels.

- The core advantage of AdaRank lies in its ability to overcome the limitations of Top-$k$ truncation. What is the distribution pattern of the
learned binary mask $B$ after optimization?

- Does it still highly conform to the traditional Top-$k$ pattern, i.e., primarily selecting the $k$ components with the largest singular values, with only the value of $k$ adaptively varying across layers?

- Or can the mask $B$ genuinely perform non-contiguous, jump-wise selection of the most beneficial subset from all singular components,
thereby demonstrating that the method effectively prunes large, detrimental components while preserving small, beneficial ones?

---

> ### Author Response · Authors · 2025-11-21
> **Rebuttal for Reviewer mGGd [1/2]**
>
> We sincerely appreciate the reviewer for the insightful comments and the recognition of our work. We address the raised concerns and questions below.
>
> ## Weaknesses
>
> > **W1.** Computational Complexity and Time Overhead (TTA/SVD)
>
> We appreciate the comment. We agree that AdaRank introduces an additional TTA overhead compared to purely algebraic merging. However, we would like to clarify that this overhead is shared by many recent adaptive merging methods and is generally considered a favorable trade-off for the performance gains obtained [1,2,3,4]. Notably, we emphasize that TTA cost in AdaRank is modest thanks to its extremely efficient design, as it only introduces per-component binary masks and layer-wise coefficients as learnable components; as shown in Table **3** in our paper, the number of learnable parameters is below 0.05% of the full model, and the total TTA time is 10 and 37 mins with ViT-B-32 and ViT-L-14, respectively. We also show that AdaRank achieves significant improvement under extremely low-resource settings, such as using only **1%** of test data (see our response in **W3**).
>
> Regarding the SVD, we clarify that a one-time SVD cost is also required by all subspace-based merging methods [5,6,7]. Nonetheless, SVD can be computed efficiently due to the modern linear algebra libraries [8]. Recent works [9,10] also demonstrate that performing SVD on massive weight matrices of LLMs is computationally feasible and widely adopted.
>
> To further demonstrate scalability, we additionally applied AdaRank to merge LLaMa-3.2-3B models (Hidden dim=3072, 32 layers) fine-tuned on three generative tasks: Instruction Following, Code Generation, and Math Solving, following MergeBench [11]. Each task is evaluated with corresponding benchmarks of IFEval, MBPP, and GSM8k. Tables **A** and **B** summarize the results: AdaRank consistently improves performance across all tasks and outperforms AdaMerging. This confirms the applicability of AdaRank to models with large dimensions and parameters, still introducing only a small amount of learnable parameters (**0.015%**). Moreover, as reported in Table **B**, the SVD cost remains marginal (**~8min**), and the entire TTA procedure finishes within **2 hours** on a single GPU. Given that (i) these costs are incurred only once before deployment, and (ii) they are negligible compared to training or fine-tuning costs, **we believe the overhead is reasonable and does not hinder practical application.**
>
> Table **A**. Performance of AdaRank on merging LLaMa-3.2-3B models.
> |  |  | Instruction (IFEval) | Math (GSM8k) | Code (MBPP) |
> |---|---|:---:|:---:|:---:|
> |Individual (Finetuned) | Instruction | **42.7** | 24.2 | 35.0 |
> ||Math| 22.2 | **68.8** | 27.4 |
> ||Code| 18.5 | 24.0 | **39.6** |
> |Merged|TA| 37.3 | 51.8 | 40.8 |
> || TA+AdaMerging | 39.3 | 53.8 | 41.6 |
> || TA+**AdaRank** | **40.2** | **55.7** | **42.2** |
> || CART | 40.9 | 57.0 | 40.8 |
> || CART+AdaMerging | 40.8 | 58.1 | 39.8 |
> || CART+**AdaRank** | **41.7** | **59.7** | **42.0** |
>
> Table **B**. Adaptation-Time Cost of AdaRank on merging LLaMa-3.2-3B models.
> ||Total Parameters|Learnable Parameters|SVD Time|TTA Time|
> |---|:---:|:---:|:---:|:---:|
> |AdaMerging | 9.63B |765|-|98.5min|
> |AdaRank|9.63B|1462272 (0.015%)|~7.6min|110.2min|
>
> > **W2.** Validity of Gradient Approximation by using Straight-Through Estimator
>
> While we acknowledge that STE is an approximation, it is the widely accepted standard for discrete optimization, validated by foundational works [12, 13, 14]. During the optimization process, we observed a smooth and stable convergence, which confirms that STE provides sufficiently accurate gradients to effectively optimize our binary masks.

---

> ### Author Response · Authors · 2025-11-21
> **Rebuttal for Reviewer mGGd [2/2]**
>
> > **W3 / Q1.** Dependence on Unlabeled Test Data and Generalizability
>
> We appreciate the comment. While it is true that test-time adaptation requires access to unlabeled test data, we argue that (i) AdaRank remains highly effective even in extremely resource-constrained settings and (ii) it can always converge back to algebraic approaches (e.g., Task Arithmetic and TSV-M) when no test data is available.
>
> To demonstrate this, we present results by varying the amount of available test data in Tables **B** and **C**. With access to only 1% of the test set, AdaRank already outperforms AdaMerging, which uses the full test dataset. Furthermore, AdaRank with 1% test data still offers significant gains over the baseline models, improving Task Arithmetic by **10.8%** and CART by **3.5%**.
>
> This effectiveness arises because AdaRank does not learn new representations but only decides which existing singular directions should be activated or suppressed. Since each singular component already isolates a task-specific update, the entropy objective only needs to identify a small, structured subset of beneficial directions, avoiding overfitting and allowing the optimization to converge reliably with very few samples.  Also, since the parameters of AdaRank are always initialized to the best-performing SVD-based configuration (e.g., TSV-M, CART) reported in prior work, AdaRank can always converge back to these methods when there are no available test samples.
>
> The results show that AdaRank remains **highly effective even in extremely resource constrained settings**, demonstrating its practicality in broad applications. We appreciate the reviewer’s comment and will include the results in the main paper.
>
> Table **B**. Performance of AdaRank on different amount of test data ratio, on ViT.
> |ViT-B-32|  | 0% (Static) | 1% | 5% | 10% | 50% | 100% | AdaMerging(100%) |
> |:---:|:---:|:---:|:---:|:---:|:---:|:---:|:---:|:---:|
> | 8 Tasks | TA+AdaRank | 69.2 | 81.2 | 84.2 | 85.2 | 87.2 | 87.9 | 80.1 |
> ||CART+AdaRank | 65.4 | 76.8 | 78.2 | 79.0 | 81.3 | 82.1 | 76.7 |
> | 14 Tasks | TA+AdaRank | 61.0 | 70.0 | 71.3 | 73.0 | 76.7 | 81.4 | 69.2 |
> |  | CART+AdaRank |84.7|86.1|87.1|87.7|88.6|89.2|85.9|
> | 20 Tasks | TA+AdaRank |79.5|82.8|84.4|85.3|85.8|86.2|82.3|
> |  |CART+AdaRank|76.8|82.7|84.3|85.2|85.8|86.4|82.7|
>
> Table **C**. Performance of AdaRank on different amount of test data ratio, on RoBERTa / GPT-2.
> |  |  | 0% (Static) | 1% | 5% | 10% | 50% | 100% | AdaMerging(100%) |
> |:---:|:---:|:---:|:---:|:---:|:---:|:---:|:---:|:---:|
> | RoBERTa | TA+AdaRank | 0.6718 | 0.6774 | 0.6909 | 0.6937 |0.6947|0.7032| 0.6762 |
> |  | CART+AdaRank | 0.6997 | 0.7197 | 0.7087 | 0.7497 | 0.7453 |0.7547|0.6954|
> | GPT-2 | TA+AdaRank | 0.5910 | 0.6330 | 0.6270 | 0.6304 | 0.6337 | 0.6328 |0.5994|
> |  | CART+AdaRank | 0.6185 | 0.6524 | 0.6484 | 0.6528 | 0.6511|0.6587 |0.6207|
>
> ---
> ## Questions
>
> > **Q2/3/4.** Distribution Patterns of Learned Binary Mask
>
> We kindly remind the reviewer that Figure 10 in the appendix already visualizes the learned masks for several MLP layers as heatmaps over singular indices. Within each task, the preserved components form a **highly non-contiguous pattern**: many top singular components are pruned, while a scattered subset of mid and bottom components are retained. The patterns differ substantially between early and late layers and across tasks, reflecting **task-specific interference structure** rather than a simple top-k.
>
> ---
> [1] Yang et al. "Adamerging: Adaptive model merging for multi-task learning." ICLR 2023.
>
> [2] Yang et al. "Representation surgery for multi-task model merging." ICML 2024.
>
> [3] Tang et al. “Merging multi-task models via weight-ensembling mixture of experts.” ICML 2024
>
> [4] Lu et al. “Twin-merging: Dynamic integration of modular expertise in model merging.” NeurIPS 2024.
>
> [5] Choi et al. “Revisiting Weight Averaging for Model Merging” arXiv 2024.
>
> [6] Gargiulo et al. “Task singular vectors: Reducing task interference in model merging.” CVPR 2025.
>
> [7] Marczak et al. “No Task Left Behind: Isotropic Model Merging with Common and Task-Specific Subspaces.” ICML 2025.
>
> [8] Kramer, Oliver. "Scikit-learn." Machine learning for evolution strategies. Cham: Springer International Publishing, 2016. 45-53.
>
> [9] Meng et al. "PiSSA: Principal Singular values and Singular vectors Adaptation of Large Language Models." NeurIPS 2024.
>
> [10] Yuan et al. "ASVD: Activation-aware Singular Value Decomposition for Compressing Large Language Models." arXiv, 2023.
>
> [11] He et al. “MergeBench: A Benchmark for Merging Domain-Specialized LLMs”, NeurIPS 2025.
>
> [12] Courbariaux et al. "Binaryconnect: Training deep neural networks with binary weights during propagations." NeurIPS 2015.
>
> [13] Hubara, Itay, et al. "Binarized neural networks." NeurIPS 2016.
>
> [14] Van et al. "Neural discrete representation learning." NeurIPS 2017.

---

### Author Response · Authors · 2025-11-22
**Revision Summary**

We thank the reviewers for their thoughtful and constructive feedback. We have revised our manuscript to reflect these suggestions. The updates include:

- Moved the adaptation-time cost analysis from the Appendix to the main text.

- Included additional experiments on loss changes across different architectures (RoBERTa, GPT-2) to demonstrate generalizability (Appendix D.1, Figure 6).

- Added extended theoretical and empirical analyses on the SVD of task vectors and interference mechanisms (Appendix D.1, Figures 7 & 8).

- Added baseline results for NLP Tasks (RoBERTa, GPT-2)

We welcome any further questions to clarify our work.

---

### Meta-Review · Area_Chair_yuoU · 2026-01-07

**Summary:**

Overall, this paper is interesting and has a very clear structure, including convincing motivations and practical solutions. Specifically, this paper introduces AdaRank, a framework that enhances SVD-based model merging by moving beyond heuristic rank selection. Then, the authors demonstrated that the fundamental assumption of top-k low-rank approximation shared in previous methods can be suboptimal due to the nature of multi-task loss. To address this, the proposed method adaptively prunes the singular components via test-time adaptation on unlabeled data, to build a task-specific subspace with low inter-task interference.

**Reviewer Concerns:**

There are several concerns in the reviews, yet, the concerns are minor ones. All the reviewers acknowledge the contributions of this paper in the field.

Reviewer mGGd has concerns about computational overhead, gradient approximation issues and issues introduced by the unlabeled test data. However, this overhead is shared by many recent adaptive merging methods and is generally considered a favorable trade-off for the performance gains obtained. For the remaining two, they are common practice in the TTA field, which should not be a reason to reject this paper. The authors made efforts to further enhance the paper for the remaining two as well.

The main concerns from Reviewer BUd9 is that only single dynamic merging method is compared. Other concerns are more like add-on suggestions. The authors have addressed the main concern by adding more experiments, which is convincing.

Reviewer o9CP raised two major concerns: technical novelty (not the paper novelty) and theoretical insights. However, like what this paper claimed, this paper mainly found the motivation and proposed an effective solutions. Finding research questions is always important, and I agree with the authors that the novelty and sigificance of this paper are still acceptable for ICLR. For the theoretical insights, the authors and the reviewer did a good job in discussion. In the end, more concise insights are concluded.

Reviewer cS7t has some questions on this paper, instead of critical concerns, and asks more experiments. The authors addressed all the concerns (some shared with other reviewers on adding experiments).

**Reviewer Scores:**

All the critical concerns are factually addressed. Thus, the final score of this paper should be at least 6,6,6,6, which is a clear acceptance for me.

---

### Decision · Program_Chairs · 2026-01-26

Accept (Poster)